# TOWARD A SCIENTIST AGENT:
# LEARNING TO VERIFY HYPOTHESES

## ABSTRACT

In this paper, we formulate hypothesis verification as a reinforcement learning problem. Specifically, we aim to build an agent that, given a hypothesis about the dynamics of the world can take actions to generate observations which can help predict whether the hypothesis is true or false. Our first observation is that agents trained end-to-end with the reward fail to learn to solve this problem. In order to train the agents, we exploit the underlying structure in the majority of hypotheses – they can be formulated as triplets (pre-condition, action sequence, post-condition). Once the agents have been pretrained to verify hypotheses with this structure, they can be fine-tuned to verify more general hypotheses. Our work takes a step towards a "scientist agent" that develops an understanding of the world by generating and testing hypotheses about its environment.

## 1 INTRODUCTION

In fields of natural sciences (physics, biology etc.), we follow scientific methods – building and testing hypotheses to develop an understanding of the world. Many classical approaches to artificial intelligence attempted to mirror this process (Brachman & Levesque, 2004; Davis & Marcus, 2015), building (symbolic) knowledge representations about the world that allow the making and testing of hypotheses. However, this process bears little resemblance to the way in which current machine learning (ML) systems learn. Both traditional IID and interactive learning settings use a single user-specified objective function that codifies a high-level task, but places no constraint on the underlying knowledge formed about the environment. In standard ML approaches, particularly those based on deep learning, any representation of the world is embedded in the weights of the model, and there is no explicit mechanism for formulating or testing hypotheses.

In this paper we take a modest step towards combining the classical approaches with the successes of modern ML to build a "scientist agent". When fully realized, such agent would be able to both make and test hypotheses about its environment. In this work we focus on the latter. Unlike standard supervised problems, there is no standard formulation, and no benchmarks or environments for hypothesis verification in interactive environments. A key contribution of our paper is framing the problem of hypothesis verification and presenting a feasible formulation for it. Specifically, we build an agent that, given a hypothesis about the dynamics of the world, can take actions to verify if the hypothesis is true or not. We formulate hypothesis verification as joint learning of: (a) an action policy that generates observations which are relevant to verification of hypotheses and; (b) a prediction function which uses the observations to predict whether the hypothesis is true or false.

We first show that even in simple environments, agents trained end-to-end using deep reinforcement learning methods cannot learn policies that can generate observations to verify the hypothesis. To remedy this, we exploit the underlying structure of hypotheses – they can often be formulated as a triplet of a pre-condition, an action sequence, and a post-condition that is causally related to the pre-condition and actions. Using this common structure, we are able to seed our action policy to learn behaviors which alter the truth of the pre-condition and post-condition. We show that this policy can be fine-tuned to learn how to verify more general hypotheses that do not necessarily fit into the triplet structure. Thus our approach allows combining the explicit hypothesis testing of classical AI with the use of scalable statistical ML.

See videos and more at: `https://sites.google.com/view/scientistagent`

## 2 RELATED WORK

**Knowledge representation and reasoning (KRR)** (Brachman & Levesque, 2004) is a central theme of traditional AI. Commonsense reasoning (Davis, 1990; Davis & Marcus, 2015; Liu & Singh, 2004) approaches, e.g. CYC (Lenat, 1995), codify everyday knowledge into a schema that permits inference and question answering. However, the underlying operations are logic-based and occur purely within the structured representation, having no mechanism for interaction with an external environment. Expert systems (Giarratano & Riley, 1998) instead focus on narrow domains of knowledge, but are similarly self-contained. Logic-based planning methods (Fikes & Nilsson, 1971; Colaco & Sridharan, 2015) generate abstract plans that could be regarded as action sequences for an agent. By contrast, our approach is statistical in nature, relying on Reinforcement Learning (RL) to guide the agent.

Our approach builds on the recent interest (Mao et al., 2019; Garcez et al., 2012) in neural-symbolic approaches that combine neural networks with symbolic representations. In particular, some recent works (Zhang & Stone, 2015; Lu et al., 2018) have attempted to combine RL with KRR, for tasks such as navigation and dialogue. These take the world dynamics learned by RL and make them usable in declarative form within the knowledge base, which is then used to improve the underlying RL policy. In contrast, in our approach, the role of RL is to verify a formal statement about the environment. Our work also shares some similarity with Konidaris et al. (2018), where ML methods are used to learn mappings from environment states to representations a planner can use.

**Cognitive Development:** Empirical research on early learning (Gopnik, 2012; Kushnir & Gopnik, 2005) has shown that infants build an understanding of the world around them in ways that parallel the scientific process: constantly formulating hypotheses about how some physical aspect of the world might work and then proving or disproving them through deliberate play. Through this process the child builds up an abstract consistent causal understanding of the world. Violations of this understanding elicit surprise that can be measured by researchers (Spelke et al., 1992).

**Automated Knowledge Base completion:** This work is also related to knowledge base completion (Fader et al., 2011; Bordes et al., 2013; Suchanek et al., 2007), and especially as formulated in (Riedel et al., 2013). Instead of using other facts in the knowledge base or a text corpus to predict edges in the KB, here the agent needs to act in an environment and observe the results of those actions. This recalls (Mitchell et al., 2018), where the system verifies facts it has previously hypothesized by searching for corroboration in the corpus.

**Automation of the scientific process** has been attempted in several domains. Robotic exploration of chemical reactivity has been demonstrated (Granda et al., 2018) using ML techniques. (King et al., 2009) developed a robot scientist that explored geonomics hypotheses about yeast and experimentally tested them using laboratory automation. In biochemistry (Vanlier et al., 2014) used Bayesian methods for optimal experiment design. More generally, the Automated Statistician project (Steinruecken et al., 2019) uses a Bayesian approach to reason about different hypotheses for explaining the data, with the aims of creating interpretable knowledge.

**Embodied Question and Answering:** The problem studied in this paper is closely related to the embodied visual question-answering problem in (Das et al., 2018). Indeed, our basic formulation is a particular case of the most general formulation of embodied QA, as the agent is rewarded for successfully answering questions about the environment that require interaction. However, the form of the questions is different than those considered in (Das et al., 2018), as they may require drawing a conclusion about the *dynamics* of the environment, rather than a static property. Even the questions about static properties we are interested in have a different flavor, as they encode rules, rather than statements about the current configuration. Our approach is built around hypothesis-conclusion structure special to these questions.

There is also a large body of work on (non-embodied) visual QA (Kafle & Kanan, 2017; Wu et al., 2016a) and text-based QA (Rajpurkar et al., 2018). From this, most relevant to our work is (Wu et al., 2016b) who use a structured knowledge base to augment standard statistical QA techniques.

**Language grounding:** Our approach requires us to solve the language grounding problem, albeit in a simplified form due to templated language/limited vocabulary. Most other works such as (Chaplot et al., 2018; Anderson et al., 2018; Tellex et al., 2011) are focused on instruction following in known or unknown environments.

**Learning to experiment:** Recent works have studied training agents to interact with an environment to draw conclusions about its dynamics (Denil et al., 2016) or elucidate its causal structure (Dasgupta et al., 2019). Our work is similar to these (especially (Denil et al., 2016) which uses reinforcement learning on sequences of observations) in that the agent gets reward for answering questions that require experimentation with the environment. However, in those works, the "question" in each environment is the same; and thus while learning to interact led to higher answer accuracy, random experimental policies could still find correct answers. On the other hand, in this work, the space of questions possible for any given environment is combinatorial, and random experimentation (and indeed vanilla reinforcement learning) is insufficient to answer questions.

## 3 PROBLEM

### 3.1 THE HYPOTHESIS VERIFICATION PROBLEM

Here we formally introduce the problem of hypothesis verification as a Partially Observable Markov Decision Process (POMDP).

The agent is spawned in an environment $W \in \mathcal{W}$ defined by the "rules" of the particular instance $W$ out of all possible worlds $\mathcal{W}$. For instance, in a crafting world, $W$ will be defined as a set or rules for what items can be crafting from which other items, and this ruleset will be different from other environments in $\mathcal{W}$

Given the environment $W$, the agent is given a hypothesis to test $h$ which relates to the rules of the world instance. By construction, $h$ is either true or false. The agent can take actions $a \in A$ (for example, move left, move right, craft, etc), including two special actions $ans_T$ and $ans_F$. The goal of the agent is to correctly identify the hypothesis $h$ as true or false and take the corresponding answering action. At the end of the episode, the agent is told whether the $h$ was true or false.

In our experiments, we set the probability of $h$ being true at $0.5$, and construct the environments such that it is not obvious from time $t = 0$ whether the hypothesis is true or not. The agent must therefore learn a hypothesis conditioned policy $\pi(s, h) : (\mathcal{S}, \mathcal{H}) \to A$, such that the agent has enough information to know whether $h$ is true.

Because we also have access to the ground truth for whether the hypothesis is true, we can train a network with supervised learning to predict true and false. Our prediction network $f(s_t, s_{t-1}, ... s_{t-N}, h) : (\mathcal{S}^N, \mathcal{H}) \to h_{pred}$ takes in the last $N$ observed observations of the environment and the hypothesis and predicts whether or not the hypothesis is true. The special $ans$ action replaces the earlier $ans_T$ and $ans_F$, and the prediction network is used to decide whether the agent answer true or false.

This addition of a supervised component is not strictly necessary for the definition of the problem. However, this framing allows for the use of supervised learning for the actual ground truth prediction, which is known to be an easier problem than the indirect optimization of RL. Empirically, this change makes the problem much more tractable.

Now, to train the policy network $\pi$, we can now define a reward function to allow for standard RL training of the policy. In essence, we give the agent a positive reward at the end of an episode if it correctly guesses the correct truth value of the hypothesis $h$.

$$R_{ans} = \begin{cases} +C & a = ans \ \& \ h_{pred} = h_{gt} \\ -C & a = ans \ \& \ h_{pred} \neq h_{gt} \\ 0 & otherwise \end{cases}$$

where $C \in \mathcal{R}^+$ is some constant reward value and $h_{gt}$ is the ground truth value of the hypothesis.

Note that any particular choice of $W$ forms an MDP if it were to be played repeatedly; but with the dynamics depending on the ruleset, the state is not fully observed, and naive RL is not applicable. As is standard in these situations, we use models that take as input a sequence of observations.

This dual optimization of policy and hypothesis prediction makes hypothesis verification a quite challenging problem! In order to be able to tell whether a hypothesis is true or not, we need to take the correct sequence of actions related to the hypothesis. But in order to know that a particular sequence of actions was the right one to do, we need to be able to correctly predict the hypothesis to know that

we should have a positive reward for that sequence. Guessing with no information gives average $0$ reward, and until it learns good predictor it has no signal to guide the policy to do the right thing. We find that a RL baseline finds it almost impossible to solve the task as it can neither learn the right policy nor the right predictor to verify the hypothesis.

## 3.2 ENVIRONMENTS

We create three games in order to test the problem of hypothesis verification: Color Switch, Pushblock, and Crafting. See Figure 1. Each instantiation of an environment comes with a hypothesis for the agent to verify. The hypotheses are generated along with the environment using a set of templates associated to the game (see Appendix A). For each spawn of the environment, the locations of the agent, all items and entities, the given hypothesis to verify, as well as the underlying logic of the world is randomized. This prevents the agent from learning the truth of the hypothesis by simply guessing without interacting with the world.

In the **Color Switch** environment, the agent is placed in a world with one or more color switches which are randomly either "on" or "off" and a door which is either open or closed. The agent is able to move and toggle the switch positions. One of the switches in the environment, when in the correct position (can be either on or off) will cause the door to open. The other switch has no effect. Hypotheses in this environment relate to the color and position of switches and how that opens or closes the door.

In the **Pushblock** environment, the agent is placed in a world with a block which can be pushed by the agent, and a door. The agent can move and push on the block. The door opens when the block is in a particular part of the grid: "up" – top two rows, "down" – bottom two rows, "left" – leftmost two rows, "right" – rightmost two rows. The hypotheses in this environment related to the position of the pushblock and how that affects the door.

Finally, in the **Crafting** environment, the agent is placed in a world with crafting rules similar to that of the popular Minecraft game. The agent is spawned along with a number of crafting items, and a crafting location. The agent is able to move, pick up items into its inventory and use the crafting location using special crafting actions. There is some true "recipe" which produces some new item in the agent's inventory.

Items are randomly generated in a $5$ by $5$ grid. The world observation is given by a 1-hot vector of each possible item in the world at each grid location and another 1-hot vector for each item and whether it is in the agent's inventory. The hypothesis is encoded as sequence of tokens. As we describe in Section 3.1, the (sparse) reward function for these environments is $C = 10$ if the agent takes the special $ans$ action and correctly verifies the hypothesis as true or false, and $-10$ if it incorrectly guesses.

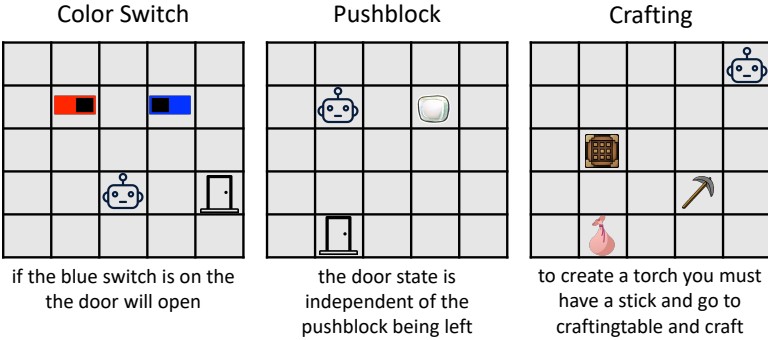

| Color Switch | Pushblock | Crafting |
| --- | --- | --- |
| if the blue switch is on the the door will open | the door state is independent of the pushblock being left | to create a torch you must have a stick and go to craftingtable and craft |

Figure 1: Examples of the Color Switch, Pushblock and Crafting hypothesis verification environments

## 3.3 HYPOTHESIS CONSTRUCTION

In the following sections, we discuss different types of hypotheses about the environment in order of increasing complexity.

### 3.3.1 TRIPLET HYPOTHESES

In the first case, we consider hypotheses that have the following "triplet" form.

$$(\text{pre-condition}, \text{action sequence}) \implies \text{post-condition}$$

The idea here is that we want to explicitly form the hypothesis as a logical statement. When the pre-condition is true, and the action sequence is performed, the post-condition will be true.

To generate our triplet hypotheses, we: (1) randomly select a pre-condition template from a set list; (2) randomly select an action template; (3) randomly select a post-condition template; and (4) fill in any entities in the final template

So for example, for the Color Switch environment we might draw "if the COLOR switch is ON_OFF_SWITCHSTATE, NULL, the door will open" and then draw "blue" for COLOR and "on" for ON_OFF_SWITCHSTATE, giving us the final template: "if the blue switch is on the door will open."

In Appendix A, we show the possible templates for each of the triplets and the possible values for all of the entities for our three environments.

### 3.3.2 GENERAL TEMPLATE CONSTRUCTION

In the more general case, instead of drawing a template from the triplet form, we instead draw a single template for the hypothesis and fill in the values. For instance, in pushblock we might draw: "the door can only be opened when the pushblock is PUSHBLOCK_POSITION"
and then draw "left" for PUSHBLOCK_POSITION. These templates are more general than the triplet ones in that they need not hold to the strict triplet form, and we have no explicit labels for pre-condition, action sequence and post-condition.

### 3.3.3 SPECIAL CASE TEMPLATES

Finally, we also can draw some more difficult and general hypothesis templates. Some of these cannot be neatly fit into a triplet format by rewording, and some may not fully describe the rules of the world. Some examples of these harder templates are: (1) Negating effects (e.g. door is not open); (2) Negating conditions (e.g. switch is not on); and Independence (e.g. door independent of blue switch). See Appendix A for all of the possible templates for an environment and further details.

## 4 METHODOLOGY

### 4.1 RL BASELINE

The conceptually simplest approach to solving the problem is to give an RL agent a sequence of $N$ observations of the form $(o_i, h)$, where $h$ is the hypothesis about the environment, and $o_i$ is the observation. As long as $N$ is large enough, a standard RL algorithm has the capacity to solve the problem.

Thus, we design our policy network $\pi(s, h)$ to decide the action. We also use the simplification described in Section 3.1 and create another network to predict the hypothesis ground truth value trained using supervised learning. The specifics of the networks are further described in Section 4.3 and hyper-parameters are described in the Appendix.

### 4.2 TRIPLET POLICY PRETRAINING

Rather than try to rely on general RL methods, we use the special structure of many hypotheses. As we discussed in Section 3.3.1, many hypotheses naturally take the form of a triplet: (pre-condition, action sequence, post-condition). While not all hypotheses fit into this format, the hope is that the policy we learn is close enough to ground truth, that we can later generalize to other kinds of hypotheses.

We can use this to construct a reward function. We know that to verify these kinds of statements, we need to take actions which alter the truth of the pre-condition and post-condition. If we modify the

pre-condition and take the action, if the statement is true, the post-condition should toggle from false to true in the environment. Similarly, if post-condition changes but the pre-condition did not change, we know that the statement must be false.

Thus we construct the following reward function to encourage our agents to toggle the pre-conditions and post-conditions:

$$R_{pre} = \begin{cases} +C & a = ans \ \& \text{ pre changed in last } N \text{ frames} \\ 0 & otherwise \end{cases}$$

$$R_{ppost} = \begin{cases} +C & a = ans \ \& \text{ post+pre changed in last } N \text{ frames} \\ 0 & otherwise \end{cases}$$

It encourages the policy to change the pre-condition and post-conditions (via pre-condition) in the last $N$ frames of the video, so that a predictor looking at the last $N$ frames of observations will be able to deduce the truth value of the hypothesis.

Once we have trained the policy function with this proxy reward, we can then train the prediction network and even finetune our policy network on the final reward.

### 4.3 NETWORK ARCHITECTURE

Although other works such as Chaplot et al. (2018) have investigated language-conditioned RL (usually in the form of instruction following), our hypothesis conditioned problem proved to be quite challenging, and required some novelty in network architectures.

For the policy networks, standard architectures were not effective for our problem. The key seems to be that it is difficult to condition action on language without explicit interaction between the language and non-language components. In particular, of all of the network architectures we experimented with, an explicit attention network using the language as the key input was by far the most effective. The hypothesis is fed into a seq2vec model and used as the key to the a dot-product attention mechanism. The state of the network (the grid locations of the items in the world and the inventory of the agent) after being fed through a one layer networks is fed as input to $N$ parallel MLPs. The output of the MLPs are fed as the values into the attention mechanism. The output of the module is then fed into the final hidden layer of the actor-critic network.

For the prediction network, we use the popular transformer architecture Vaswani et al. (2017). Our prediction network encodes both the hypothesis and past observations (after they are passed through a one layer network) using transformer encoders. These sequences are then combined using a transformer to generate a final hidden state as output which is then fed to a final prediction layer and sigmoid function to get our binary prediction.

In Figure 5, we provide ablation analysis for both of our neural network architectures. See Appendix C for more network details and hyperparameters and network diagrams.

## 5 EXPERIMENTS

First, we train using our policy networks using our pretraining proxy functions from Section 4.2. We find that pretraining with just the pre-condition reward leads to better results for the Color Switch environment, and use both rewards for the other two environments. Figure 2 shows these results.

Next, we train our network on the final prediction reward and train our prediction networks. We train two different versions of this. For one, we only train the prediction network and keep our policy network fixed. For the other, we train both the prediction network and finetune the policy network.

During this final training stage, we relax our triplet-form constraint and train on both the triplet-templated hypotheses we saw during pretraining as well as new hypothesis templates not seen during pretraining. We sample seen versus new templates with equal probability. See Section 3.3 and Appendix A for examples of the kinds of hypotheses we see during this phase of training. Note that this includes hypotheses which break the triplet format.

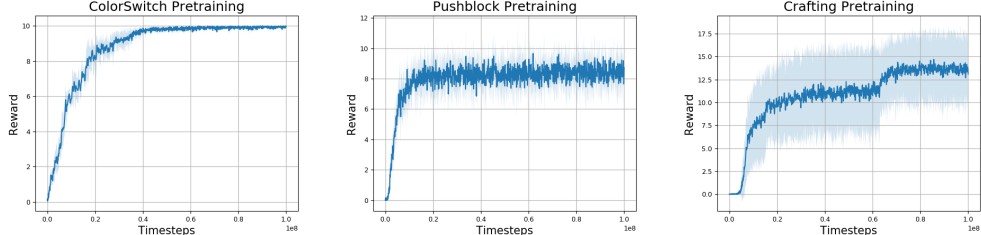

Figure 2: Pre-condition Post-condition reward pretraining on our verification environments

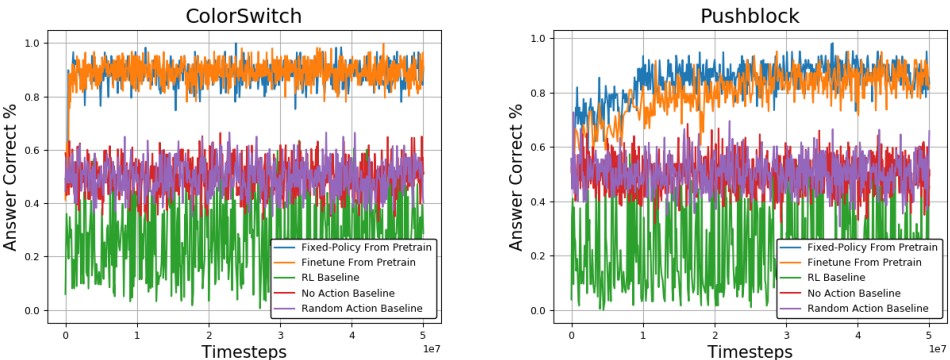

Figure 3: Final accuracy on both triplet and non-triplet hypotheses on Color Switch (left) and Pushblock (right)

Figure 3 and left of Figure 4 show our final hypothesis verification results. We show the max out of five for each of the methods shown. We also break down the final hypothesis prediction accuracy for our methods in Table 1, and show its success on the triplet hypotheses (which our methods were pretrained on) and non-triplet hypotheses (which they were not).

**RL baseline** We can first see clearly that the RL baselines fail. This is due to the unlikelihood of taking the right actions to verify correctly and therefore train the prediction net properly. Because the average reward for answering is 0 if you cannot predict correctly, the agent does not even bother answering the question much of the time (which is why this baseline gets less than 50%, it does not bother guessing in most games).

**Other baselines** We also include two other simple baselines "no act" and "random act." The no act baseline simply takes the $ans$ action at $t = 0$ and the prediction network attempts to predict the hypothesis with just the first observation. This fails because the agent needs to take actions in the environment to be able to predict the hypothesis accurately. For random act, we simply make the policy to take random actions. This similarly fails as random actions are extremely unlikely to behave in a way that allows for the verification of the hypothesis.

### 5.1 TRIPLET POLICIES CAN SUCCEED AND GENERALIZE

On the other hand, we see that RL is able to train on the triplet tasks after pre-training. While it is not surprising that densifying the reward in this way makes the RL easier, in our view, it is important that it is true, as it paves the way towards hypothesis verifying agents. That is: we are interested in scalable methods that can use statistical ML to interact with a complex environment. Given the more general success of deep RL, that the problem becomes approachable with reasonable reward shaping gives us hope we will be able to get beyond the regime of classical AI methods.

Morever, in Pushblock and Color Switch, even with the policy learned from the triplet pre/post reward, the agent is able to generalize and perform well on templates not seen in the pre-training phase as we can see in Table 1. This includes generalizing to difficult templates such as negations

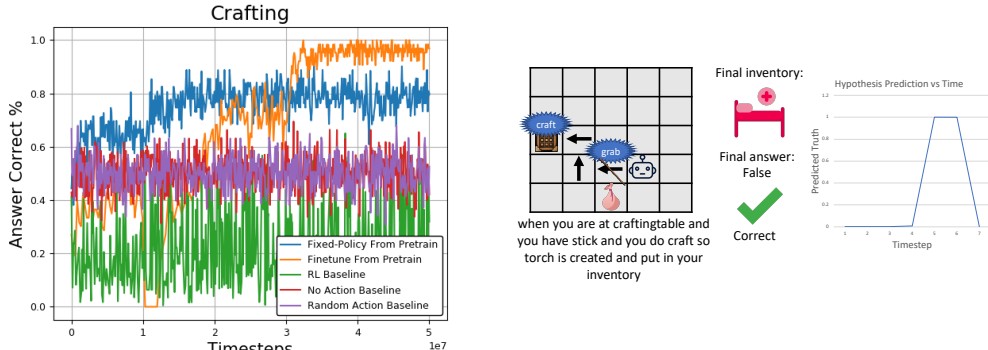

Figure 4: Final accuracy on both triplet and non-triplet hypotheses on Crafting (left) and a visualization of the finetuned policy (right)

and "independence" hypotheses. Note that the prediction network that verifies the hypotheses given the trajectory from the policy still needs to fine-tune on the new templates.

It's worth noting that although we can do well using finetuning using a few random seeds, these methods are high variance. In the appendix we show and discuss this more clearly. In Figure 7 we show the variances of these methods which show that the variance on our method is high. In Appendix E we propose a training methodology that sorts out the bad random seeds by using the triplet hypotheses as a validation set. And in Appendix J we show that these results are consistent when we increase the number of random seeds to 25.

## 5.2 TRIPLET POLICIES CAN ADAPT

On the crafting task, to do well on the unseen templates, the policy also needs to be fine-tuned. In our view, the fact that this fine-tuning can succeed is more important than the generalization in the simpler tasks, as it demonstrates a path towards agents that can verify complex statements by establishing a curriculum of simpler ones.

In the right of Figure 4, we show a visualization of a sample run of the finetuned policy and predictor on crafting. We see that the policy does what we expect, picks up the correct item and moves to the crafting table to craft. It crafts a different item than it expected (bed instead of torch) and it answers false. Looking at the prediction net over time, we see that it at first predicts false then true before it does the craft action. Once it has crafted the bed, however, it answers correctly.

Table 1: Hypothesis Prediction, broken down by triplet (pre-trained) and non-triplet (not seen in pre-training)

|  | Method | Overall | Triplet Accuracy | New Template Accuracy |
|---|---|---|---|---|
| **Color Switch** | Fixed Policy | 89.6% | 94.3% | 88.7% |
|  | Finetuned Policy | 89.3% | 92.3% | 86.3% |
| **Pushblock** | Fixed Policy | 88.1% | 89.7% | 86.5% |
|  | Finetuned Policy | 85.1% | 85.2% | 85.4% |
| **Crafting** | Fixed Policy | 79.3% | 91.4% | 69.9% |
|  | Finetuned Policy | 95.9% | 96.7% | 95.1% |

We conduct additional experiments in the Appendix. In Appendix G, we tease further analyse the problem by experimenting with an oracle hypothesis predictor. In Appendix F we experiment with different pretraining functions. In appendix Appendix H we look at training baselines for longer. And In Appendix I, we look at whether giving the baselines more past frames $N$ improves performance.

In Figure 5 we see the results of our network architecture ablation. As we can see, our new policy architecture described in Section 4.3 clearly outperforms a standard MLP policy network on the

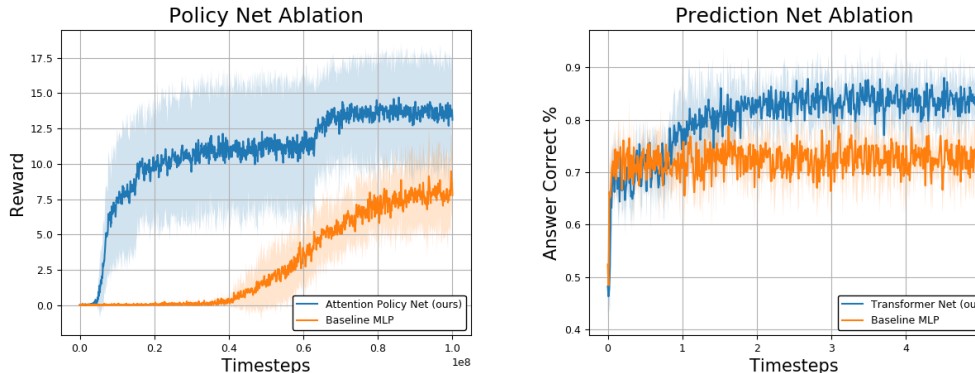

Figure 5: (left) policy network ablation (right) prediction network ablation.

language-condition pretraining task. We also see that the transformer architecture outperforms the LSTM and MLP model on the final task when we hold the policy network constant.

## 6 DISCUSSION

In this work, we propose a tractable formulation of the problem of training agents that can interact with an environment to test hypotheses about it. We show that generic RL techniques struggle with the problem, but by using its structure, we are able to develop a method that works in simple environments. Specifically, we use the fact that many hypotheses can be broken into triples of the form of (pre-condition, action sequence, post-condition); but we also show that once pre-trained using this factorization, agents can be fine-tuned to verifying more general hypotheses.

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

# A    TEMPLATES

## A.1    WORLD AND HYPOTHESIS CONSTRUCTION

Returning again to our notation from Section 3.1, the environment at each spawn needs to construct a world $W$ out of all possible $\mathcal{W}$, and a hypothesis $h$ that is either true or false in the world. $W$ in particular describes the rules about how the environment works (i.e. which switch opens the door) which in our case can precisely be describe by a hypothesis. So given a true hypothesis, we can exactly describe the rules of the world. Therefore, in order to create an instance of a possible $W in \mathcal{W}$, we can instead draw a true hypothesis about the world at random. From the hypothesis, we can then construct the rules the determine how objects in the world behave. Note that there are couple exceptions to this for our harder hypotheses, where the hypothesis can be true but only partially describes all the rules of $W$. For these cases, we draw yet another template which is consistent with the hypothesis and use that to construct the rules, such as deciding which color switch really opens the door.

Because we have to randomly give either a true or false hypothesis, we also need to be able to generate a false hypothesis for the world. So for every instance, we also draw a random false hypothesis. Now, given a true and false hypothesis, we can fully generate the world and all the items that appear in either statement. So for instance, if the true hypothesis mentions a green switch and the false one mentions a blue switch, we generate both a green and blue switch. Then, we can set the rules such that the right thing happens. So in this example, switching the green switch opens the door and the blue switch does nothing.

The final step is then to randomly choose either the true or false statement as the "visible" hypothesis which is passed to our agent to verify. Because we generate the world and spawn the items before we make this choice, we ensure that we do not accidentally give away the truth of the hypothesis based on what items spawned.

Our process for generating a new spawn of environment can thus be summarized as follows:

1. We randomly generate a true hypothesis
2. We randomly generate a false hypothesis
3. We construct a ruleset from the true hypothesis
4. We spawn the agent and the items in the world described in both the true and false hypothesis
5. We randomly choose either the true or false hypothesis as the "visible" hypothesis that the agent must verify

**Color Switch**:

**Pre-condition**:
if the COLOR switch is ON_OFF_SWITCHSTATE
when the COLOR switch is in the ON_OFF_SWITCHSTATE position
the COLOR switch is ON_OFF_SWITCHSTATE

**Action**:
""

**Post-condition**:
then the door is open
the door is passable
and we see the door is open
the door will open

**Finetune templates**:
the door can only be opened by switching the COLOR switch to ON_OFF_SWITCHSTATE
when we see the COLOR switch is ON_OFF_SWITCHSTATE the door must be open

if the COLOR switch turns ON_OFF_SWITCHSTATE the door opens
when we see the door open it must be that the COLOR switch is in the ON_OFF_SWITCHSTATE position
those who want to open the door must first switch the COLOR switch ON_OFF_SWITCHSTATE
no password just make the COLOR switch be ON_OFF_SWITCHSTATE to open the door
COLOR switch ON_OFF_SWITCHSTATE implies door is open
only the COLOR switch being ON_OFF_SWITCHSTATE opens the door
the door is open because COLOR switch is in the ON_OFF_SWITCHSTATE position
COLOR switch ON_OFF_SWITCHSTATE equals open door
the COLOR switch opens the door but only when it is ON_OFF_SWITCHSTATE
door is open must mean that COLOR switch is ON_OFF_SWITCHSTATE an ON_OFF_SWITCHSTATE means the door is
open but only if it is COLOR
COLOR controls the door and it opens when it is ON_OFF_SWITCHSTATE
ON_OFF_SWITCHSTATE is the correct position of the COLOR switch and it opens the door
the switch that causes the door to be open when it is ON_OFF_SWITCHSTATE is COLOR
if you see COLOR switch then the door is open
the door is independent of the COLOR switch
if the door is not open then the COLOR switch must be ON_OFF_SWITCHSTATE
if the COLOR switch is not ON_OFF_SWITCHSTATE then the door is open
to make the door not open the COLOR switch must be not ON_OFF_SWITCHSTATE
whether the door is open is completely independent of the COLOR switch
the COLOR switch is what controls the door
a not ON_OFF_SWITCHSTATE COLOR switch opens the door

**Template Values**
**COLOR**:
blue
red
green
black

**ON_OFF_SWITCHSTATE**:
on
off

## Pushblock

**Pre-condition**:
whenever the pushblock is in the PUSHBLOCK_POSITION
if the pushblock is at the PUSHBLOCK_POSITION
the pushblock is at the PUSHBLOCK_POSITION

**Action**:
""

**Post-condition**:
then the door is open
the door is passable
and we see the door is open
the door will open

**SP_FULL_TRAIN**:
PUSHBLOCK_POSITION is the correct position for the pushblock for the door to open
if the door is open it must be that the pushblock is at the PUSHBLOCK_POSITION
when the door is open it is because the pushblock is in the PUSHBLOCK_POSITION
when the pushblock is at the PUSHBLOCK_POSITION the door is open
pushblock PUSHBLOCK_POSITION means door open
the door can only be opened when the pushblock is PUSHBLOCK_POSITION
if the pushblock is PUSHBLOCK_POSITION it means the door is open
PUSHBLOCK_POSITION pushblock opens the door
open door implies pushblock PUSHBLOCK_POSITION

open door means pushblock PUSHBLOCK_POSITION
door opens when PUSHBLOCK_POSITION is where the pushblock is
PUSHBLOCK_POSITION is the correct position for the pushblock to open the door
the door when the pushblock is PUSHBLOCK_POSITION is open
PUSHBLOCK_POSITION position of the pushblock causes the door to open
door only opens on PUSHBLOCK_POSITION pushblock
door can only open with pushblock being PUSHBLOCK_POSITION
the pushblock being at the PUSHBLOCK_POSITION is completely independent of the door
the pushblock being PUSHBLOCK_POSITION is independent of the door being open
the door state is independent of pushblock PUSHBLOCK_POSITION
PUSHBLOCK_POSITION pushblock and door are independent

**Pushblock values**:
**PUSHBLOCK_POSITION**:
left
right
top
bottom

## Crafting

**Pre-condition**:
when you are at LOCATION and you have CRAFTING_ITEM
you are at LOCATION and have in your inventory CRAFTING_ITEM
whenever you have a CRAFTING_ITEM and are at LOCATION

**Action**:
and you do CRAFTING_ACTION
then you CRAFTING_ACTION

**Post-condition**:
you now have CREATED_ITEM in your inventory
then CREATED_ITEM is created
and this creates CREATED_ITEM
so CREATED_ITEM is created and put in your inventory
then CREATED_ITEM is made

**Finetune Templates**:
to create a CREATED_ITEM you must have CRAFTING_ITEM and go to LOCATION and do the action CRAFT-ING_ACTION
CREATED_ITEM can be created by doing CRAFTING_ACTION at LOCATION when CRAFTING_ITEM is in inventory
whenever you do CRAFTING_ACTION and have CRAFTING_ITEM at LOCATION a CREATED_ITEM is made
you have CRAFTING_ITEM and go to LOCATION and CRAFTING_ACTION and CREATED_ITEM will be created
whoever does CRAFTING_ACTION at LOCATION with CRAFTING_ITEM gets CREATED_ITEM
if you have CRAFTING_ITEM at LOCATION and you CRAFTING_ACTION you get CREATED_ITEM
if you do CRAFTING_ACTION at LOCATION with CRAFTING_ITEM you make CREATED_ITEM
whenever you have CRAFTING_ITEM at LOCATION and do CRAFTING_ACTION then you make a CREATED_ITEM
having CRAFTING_ITEM in your inventory being at LOCATION and doing CRAFTING_ACTION creates CRE-ATED_ITEM
CREATED_ITEM can be made with CRAFTING_ITEM when you do CRAFTING_ACTION at LOCATION
CRAFTING_ITEM plus LOCATION plus CRAFTING_ACTION equals CREATED_ITEM
create a CREATED_ITEM by being at LOCATION with CRAFTING_ITEM and doing CRAFTING_ACTION
CRAFTING_ACTION at LOCATION creates CREATED_ITEM but only if you have a CRAFTING_ITEM
if you want to make a CREATED_ITEM then go to LOCATION with CRAFTING_ITEM and do CRAFTING_ACTION
CRAFTING_ITEM in inventory at LOCATION makes CREATED_ITEM if you do CRAFTING_ACTION
CREATED_ITEM when CRAFTING_ITEM at LOCATION and do CRAFTING_ACTION
if you are at LOCATION and do CRAFTING_ACTION you make CREATED_ITEM
if you are anywhere and do CRAFTING_ACTION with CRAFTING_ITEM you make a CREATED_ITEM
having CRAFTING_ITEM at LOCATION and doing CRAFTING_ACTION does not make a CREATED_ITEM
CREATED_ITEM is created by being at LOCATION and doing CRAFTING_ACTION
make a CREATED_ITEM by having a CRAFTING_ITEM and doing CRAFTING_ACTION

you have CRAFTING_ITEM and go to LOCATION and CRAFTING_ACTION and CREATED_ITEM will not be created
LOCATION plus CRAFTING_ACTION creates a CREATED_ITEM
with a CRAFTING_ITEM you can make a CREATED_ITEM by doing CRAFTING_ACTION

**Template Values**:
CRAFTING_ITEM :
iron
wood
stick
pickaxe
coal

**CREATED_ITEM**:
torch
bed

**LOCATION**:
craftingtable
**CRAFTING_ACTION**:
craft

# B  LEARNING DETAILS AND HYPERPARAMETERS

One detail of the prediction network is that we need to keep a memory of past state sequences, hypotheses and ground truths so we can actually train our prediction network. We do this by simply keeping track of the last $N$ times our agent answered a question, and keeping these in a FIFO memory. When we update our prediction network, we randomly sample from this pool. This also necessitates a 100k step break in period to collect enough examples.

In our policy finetuning experiments, we also stabilize our dual optimization problem by trading of optimization of the policy network and the prediction network. We must also start with the prediction network so that the reward for answering correctly is meaningful.

Table 2: Pretraining Hyperparameters

| Parameter | Value |
|---|---|
| Algorithm | PPO (Schulman et al., 2017) |
| Timesteps per batch | 2048 |
| Clip param | 0.2 |
| Entropy coeff | 0.1 |
| Number of parallel processes | 8 |
| Optimizer epochs per iteration | 4 |
| Optimizer step size | $2.5e^{-4}$ |
| Optimizer batch size | 32 |
| Discount $\gamma$ | 0.99 |
| GAE $\lambda$ | 0.95 |
| learning rate schedule | constant |
| Optimizer | ADAM Kingma & Ba (2014) |
| Past Frame Window Size | 5 |

Table 3: Finetuning Hyperparameters

| Parameter | Value |
|---|---|
| Algorithm | PPO (Schulman et al., 2017) |
| Timesteps per batch | 2048 |
| Entropy coeff | 0.1 |
| Number of parallel processes | 8 |
| Optimizer epochs per iteration | 4 |
| Optimizer step size | $1e^{-5}$ |
| Optimizer batch size | 32 |
| Discount $\gamma$ | 0.99 |
| GAE $\lambda$ | 0.95 |
| learning rate schedule | constant |
| Optimizer | SGD |
| Past Frame Window Size | 5 |

Basis of RL implementations was from Kostrikov (2018)

Table 4: Prediction Hyperparameters

| Parameter | Value |
|---|---|
| Timesteps per batch | 2048 |
| Optimizer step size | $1e^{-3}$ |
| Optimizer batch size | 128 |
| learning rate schedule | constant |
| Optimizer | ADAM Kingma & Ba (2014) |
| Memory Burn-in | 100000 |
| Memory Size | 200 |
| Alternate Training Window | 10000000 |

## C    NETWORK DETAILS AND HYPERPARAMETERS

### C.1    RELATED WORK

Other works such as Chaplot et al. (2018) have incorporated gated mechanisms between language and perception. Manchin et al. (2019) employs self-attention mechanism within convolutional layers and Choi et al. (2017) also employs a self-attention mechanism in a DQN. Neither work incorporates language and the architectures are quite different from each other.

Figure 6 shows the policy and transformer architectures.

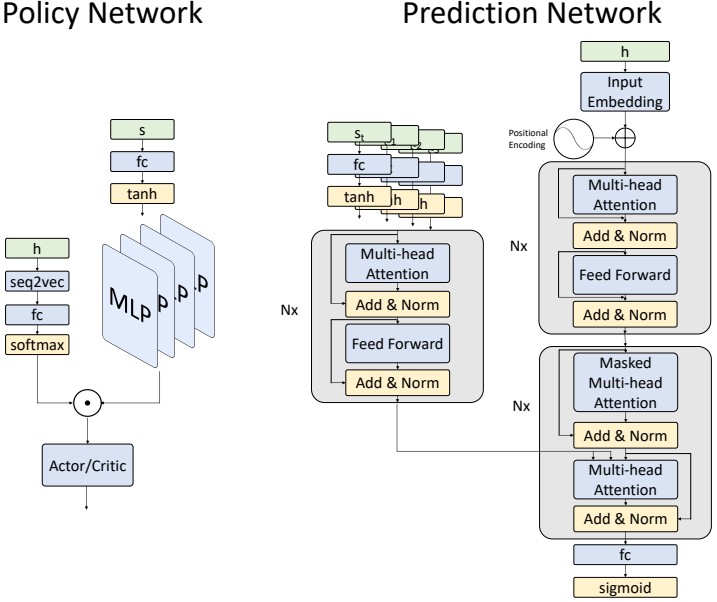

Figure 6: Network architecture for our policy network (left) and prediction network (right)

### C.2    IMPLEMENTATION AND HYPERPARAMETERS

We take much of our implementation of transformers from Rush (2018).

Table 5: Policy Network Hyperparameters

| Parameter | Value |
| --- | --- |
| Seq2Vec Model | Bag-of-Words |
| Word Embedding Size | 32 |
| Hidden Size | 32 |
| MLP Num Hidden Layers | 2 |
| Number of MLP Modules | 16 |
| Transfer Layer | $tanh$ |

Table 6: MLP Baseline Policy Network Hyperparameters

| Parameter | Value |
|---|---|
| Seq2Vec Model | Bag-of-Words |
| Word Embedding Size | 32 |
| Hidden Size | 32 |
| MLP Num Hidden Layers | 2 |
| Transfer Layer | $tanh$ |

Table 7: Transformer Network Hyperparameters

| Parameter | Value |
|---|---|
| Word Embedding Size | 32 |
| Hidden Size | 32 |
| Transfer Layer | $ReLU$ |
| Transformer $N$ | 3 |

Table 8: Baseline Prediction Network Hyperparameters

| Parameter | Value |
|---|---|
| Seq2Vec Model | LSTM |
| LSTM Num Layers | 1 |
| Word Embedding Size | 32 |
| Hidden Size | 32 |
| MLP Num Hidden Layers | 2 |
| Transfer Layer | $tanh$ |

# D ADDITIONAL FIGURES

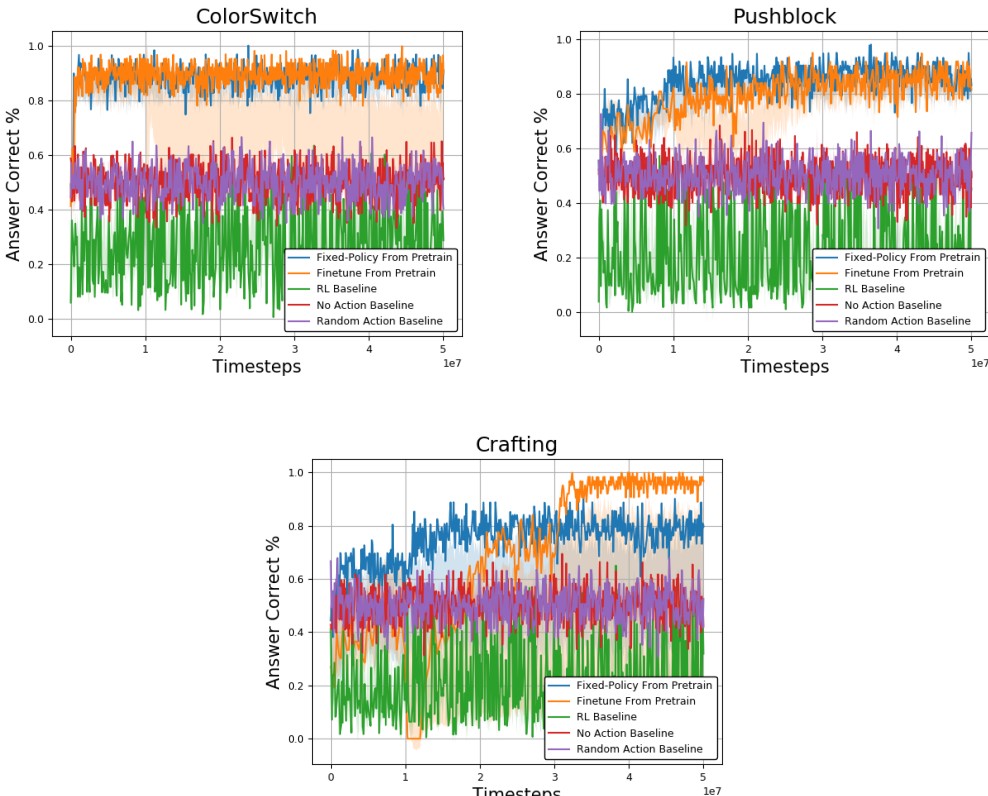

Figure 7: Final accuracy on both triplet and non-triplet hypotheses on Color Switch Pushblock and Crafting with variance plotted

# E    STAGED RANDOM SEED VALIDATION

In this experiment, we perform a two-stage procedure for evaluating our results. The idea is that we use one set of hypotheses to determine which random seeds are successful and then show results on the larger set of hypotheses.

In the first stage, we train and our methods on only the triplet templates (the same ones used during pre-training). We then choose only the seeds that performed well (in these figures we show results for keeping seeds with at least 80% prediction accuracy and with at least 90% accuracy. If a method has no seeds performing high enough, we choose the top 5 for that experiment.) We show results on 25 random seeds. We preserve all training and network hyper-parameters.

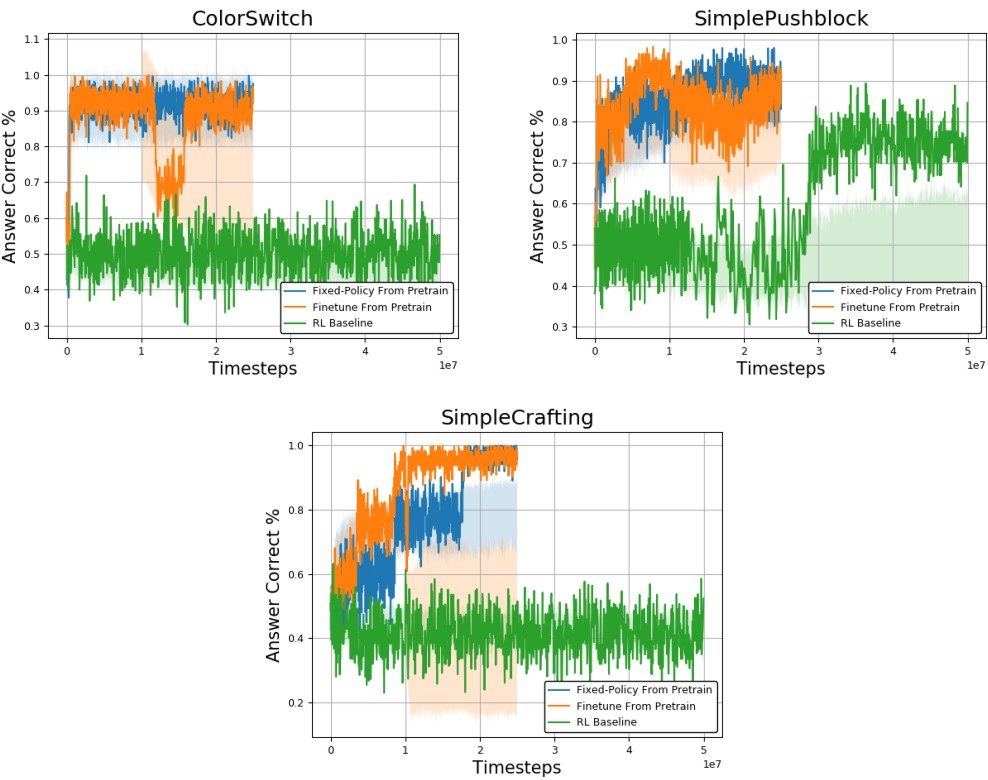

Figure 8: Hypothesis accuracy on only the triplet hypotheses for the Color Switch, Pushblock, and Crafting environments. Shown with the max seeds and the variance bands with 25 random seeds.

In Figure 8 we show the first stage of training. We only train these with the triplet templates also seen during pre-training. We give the baseline more time to train to make up for the extra time the other methods got during pretraining. We can see that for all three environments the pretrained methods have at least one good seed for both finetuning and fixed policies. For crafting, we can get a better max seed with finetuning. However, especially in crafting, the variance is quite high, with many seeds doing poorly. The baselines do poorly overall except for a couple seeds in pushblock. This is the simplest environment, so it makes sense that this would be the one where the baseline RL might be able to find a policy. The max of this still slightly underperforms the pre-trained policies.

In Figure 9 and Figure 10, we show the results in the second stage of training. As we discussed, this stage includes the more difficult, non-triplet templates not seen during pre-training and not seen during the first stage of training when we selected the top seeds. We can see that with the pruning of bad seeds, the variance bands for the pre-train methods is much smaller and more clearly outperforms the baselines. We again see that we are able to get the best results from the finetuning on crafting. As with stage 1, we see that the RL only baseline is able to do reasonably well on pushblock, but still not as good as our pre-training methods. We show results for cutoffs at 80% and 90% to make sure we were robust to the choice of cutoff, and we can see very little difference between them.

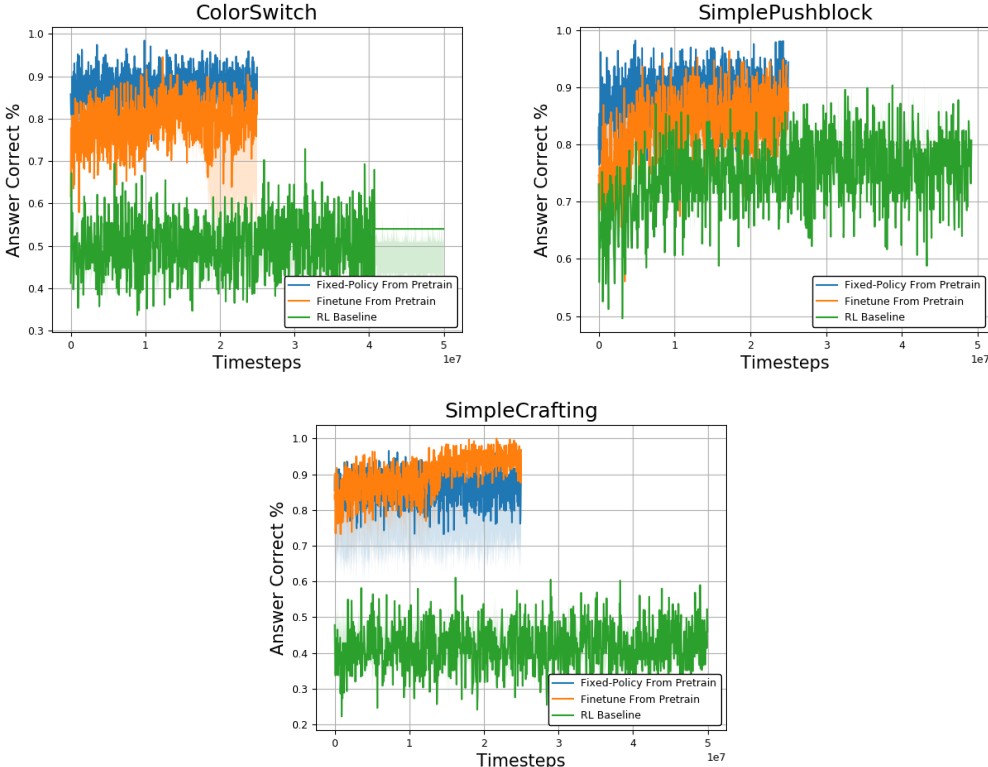

Figure 9: Hypothesis accuracy on both triplet and non-triplet hypotheses for the Color Switch, Pushblock, and Crafting environments. Shown with the max seeds and the variance bands. We use a 80% accuracy as a cutoff for this figure.

Figure 11 shows the 90% cutoff experiment again with the mean instead of the max plotted.

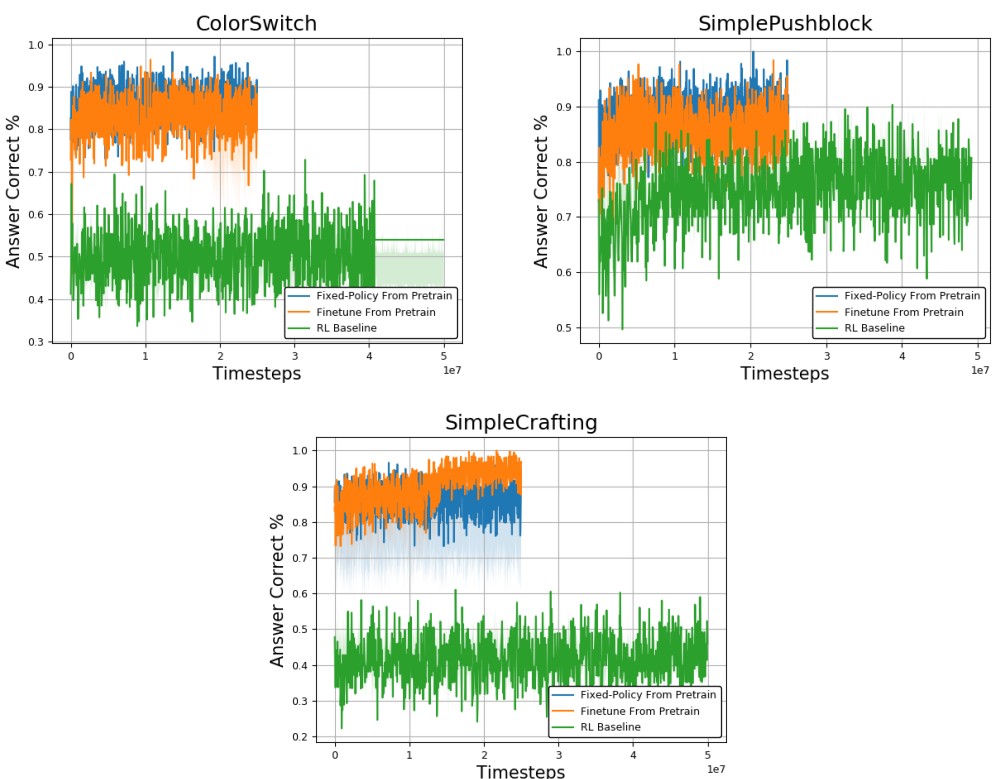

Figure 10: Hypothesis accuracy on both triplet and non-triplet hypotheses for the Color Switch, Pushblock, and Crafting environments. Shown with the max seeds and the variance bands. We use a 90% accuracy as a cutoff for this figure.

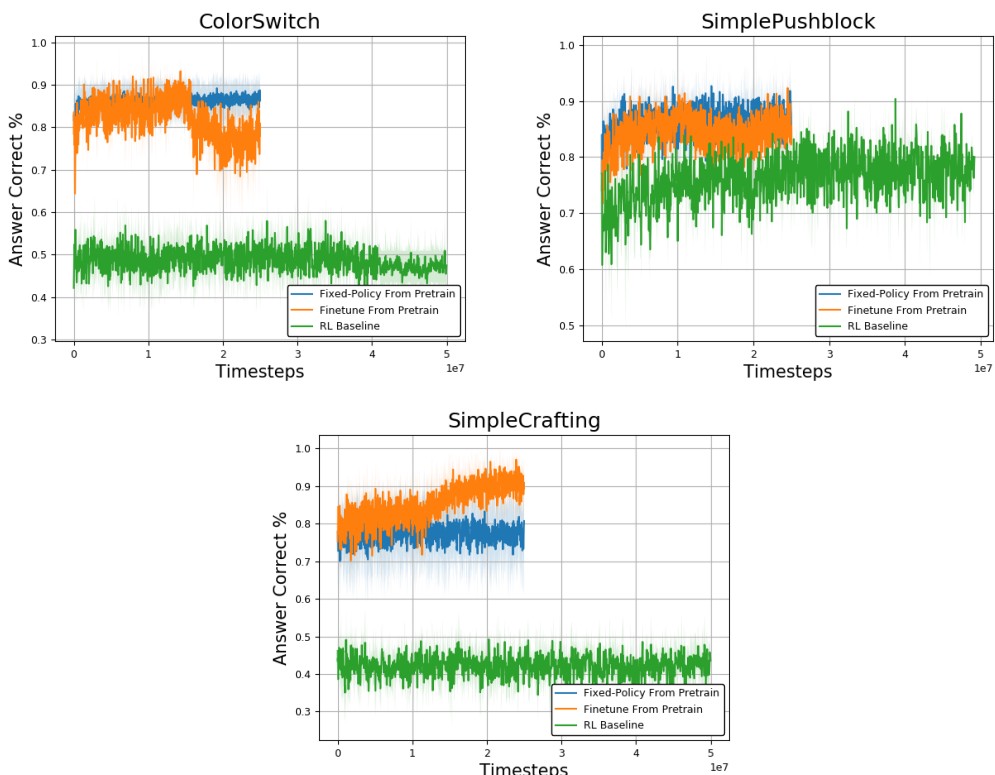

Figure 11: Hypothesis accuracy on both triplet and non-triplet hypotheses for the Color Switch, Pushblock, and Crafting environments. Shown with the means and the variance bands. We use a 90% accuracy as a cutoff for this figure.

# F  INTRINSIC PRE-TRAINING EXPERIMENTS

In this experiment, we show results on our hypotheses verification problem using different forms of "intrinsic motivation" pre-training. We show results for 4 different pretraining schemes:

1. Change any item state in the world. Receive reward at end.
2. Change any item referenced in the hypothesis. Receive reward at end.
3. Change any item state in the world. Receive reward instantaneously.
4. Change any item referenced in the hypothesis. Receive reward instantaneously.

Reward at the end means that it operates similar to our hypothesis pre-training. Specifically, the agent get reward only at the end of the episode when it has taken a stop action. At that step it gets a $+C$ reward if it changed within the last $N$ frames. For these rewards, we choose $C = 10$.

Instantaneous reward is what it sounds like. When the object state is changed, the reward is instantly received by the agent. We chose $C = 1$ for colorswitch and pushblock and $C = 5$ for crafting.

We interpret "item" to mean any object that is not the agent. So this includes crafting items, switches, pushblocks, etc. We show results on 25 random seeds. We preserve all training and network hyper-parameters.

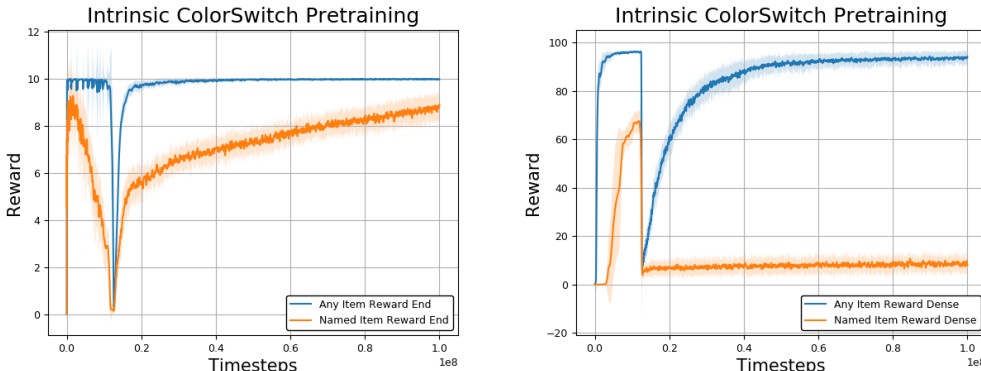

Figure 12: Pretraining Reward for ColorSwitch for intrinsic motivation. Showing mean and variance bands on 25 random seeds.

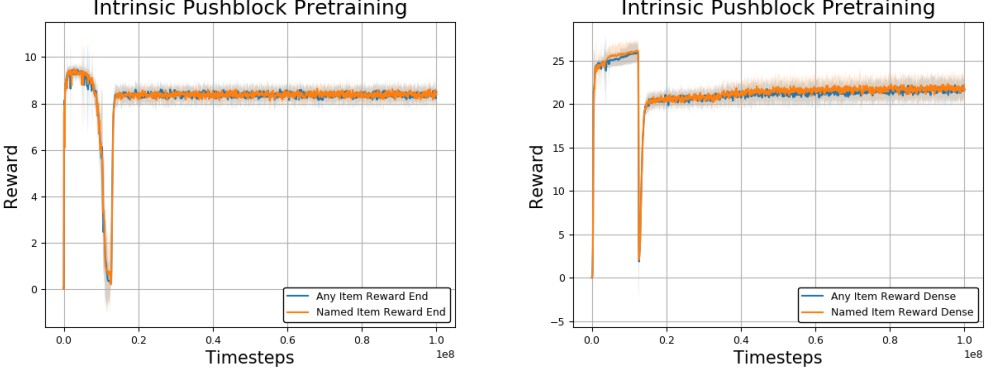

Figure 13: Pretraining Reward for Pushblock for intrinsic motivation. Showing mean and variance bands on 25 random seeds.

In Figure 15 we show the final accuracies on the hypothesis verification task using the pretrained intrinsic rewards. As before, only the hypothesis predictor and not the policy is trained at this step. In

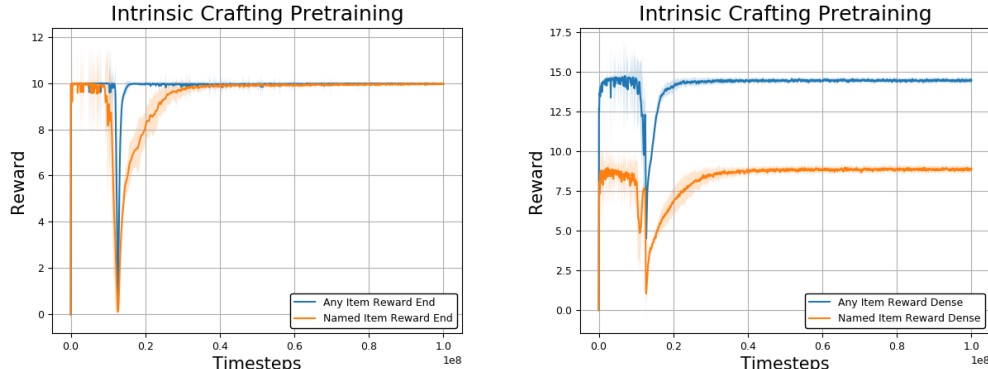

Figure 14: Pretraining Reward for Crafting for intrinsic motivation. Showing mean and variance bands on 25 random seeds.

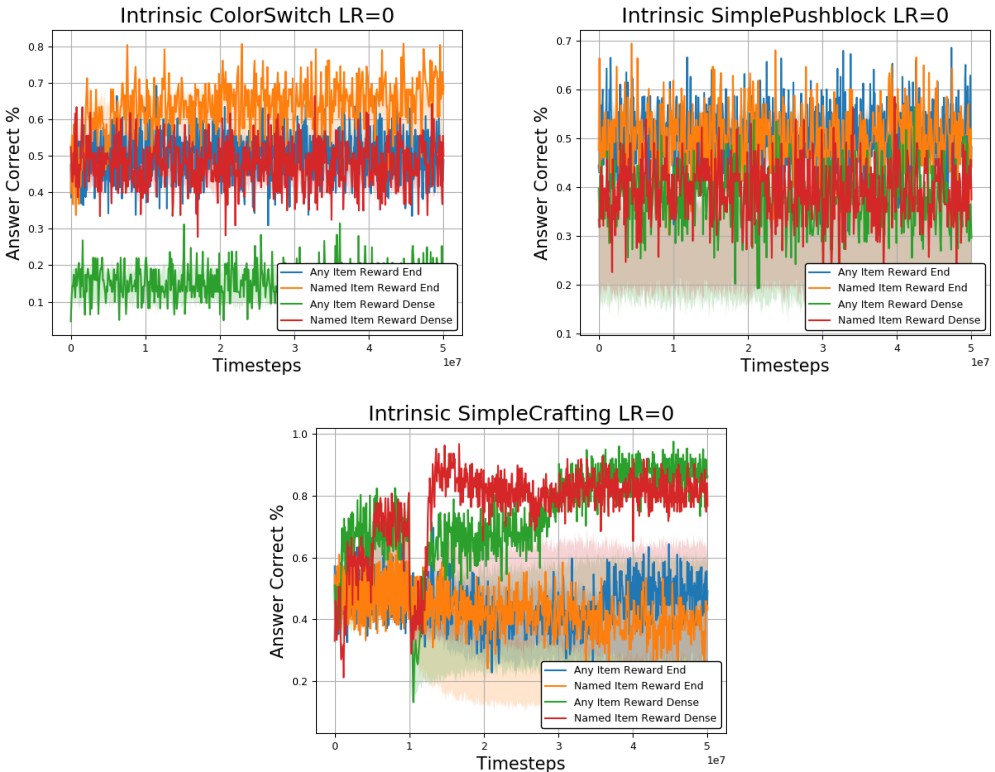

Figure 15: Final hypothesis accuracies using intrinsic pre-training. Without finetuning of policy. Showing max results and variance bounds on 25 seeds.

Figure 16 we show the same results where we finetune the policy as well. All training and network parameters are kept the same from earlier experiments.

We can see that the best results come from the crafting pre-training intrinsics. This makes a lot of sense because changing the state for crafting includes picking up objects and crafting objects, which is what the agent needs to do to verify the hypothesis. On colorswitch, we are able to get reasonable results, at least for the fixed policy. Again, changing the state corresponds to flipping switches which is also useful for verify colorswitch hypotheses. For pushbloc, nothing performed better than chance. Here, merely changing the state of the object isn't enough to verify anything. To verify pushblock hypotheses, the state of the pushblock (it's position) needs to be changed in a specific way: pushed

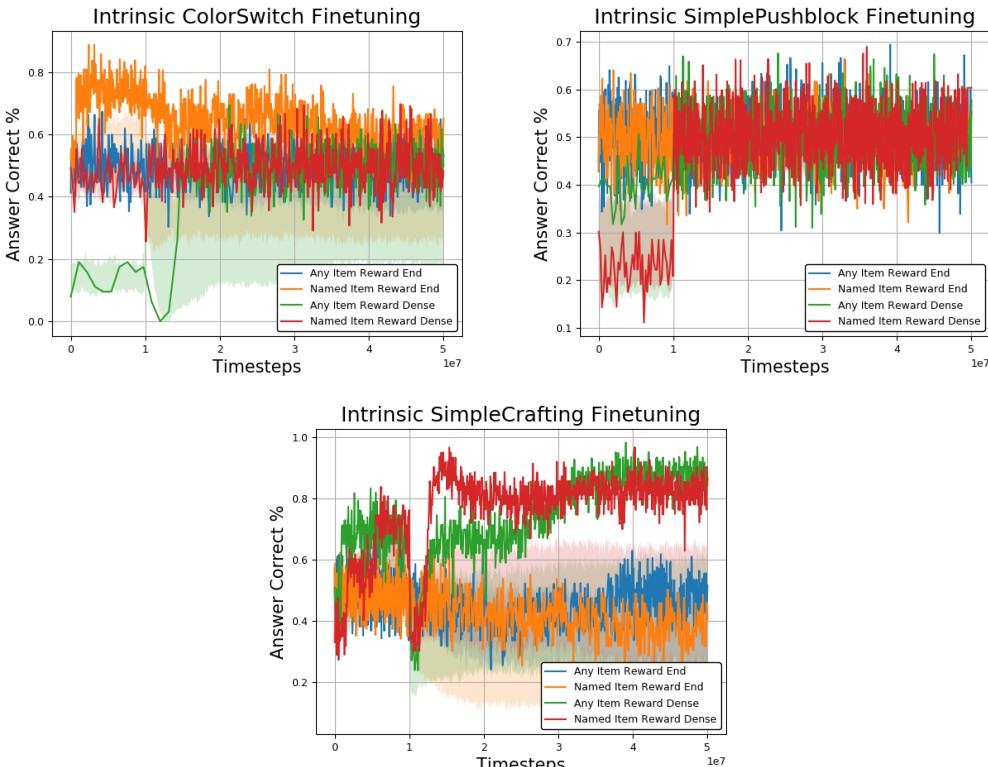

Figure 16: Final hypothesis accuracies using intrinsic pre-training. With finetuning of policy. Showing max results and variance bounds on 25 seeds.

into or out of the correct position. The intrinsic change reward does not necessarily cause this, so this did not appear to be sufficient in this case.

# G   ORACLE HYPOTHESIS PREDICTION

In this experiment, we disentangle the problem somewhat for analysis by running experiments with an "oracle" hypothesis predictor on the Crafting environment. Specifically, in these experiments, we assume that we have an oracle that, given the last $N$ states of the world, if it is possible to infer the truth state of the hypothesis given that sequence of states, the oracle returns the ground truth of the hypothesis. This should allow us to analyize the upper bounds of this problem and see what the hard part of our problem is.

First, we train a RL agent with access to the oracle. So the RL agent must learn its action policy, but when it takes the answer action, it uses the oracle to predict the hypothesis. Therefore, if the actions it has taken can verify the hypothesis, it will automatically answer correctly and get the reward. We show results on 25 random seeds and preserve the hyper-parameters from other experiments.

We show the result of this in Figure 17. We see that the RL is quickly, although not instantly, able to converge to perfect performance. From this we should summize that if we know how to predict the hypothesis already, it's quite easy to get the reward - we just have to learn to do the patterns necessary to make the oracle prediction possible. The RL baseline without pretraining without the oracle was not able to converge to a good solution. This suggests perhaps that the problem is how to get a good hypothesis predictor in the first place to let us then learn the right policy.

Toward that end, we analyze our trained algorithms to see whether the actions they take are capable of verifying the hypothesis. We show the values for the top accuracy model. We use the same models and seeds whose results we show in Table 1 and Figure 4.

Table 9: Oracle Evaluation of Learned Policies

| Method | Oracle % can answer | Theoretic Upper Bound |
|---|---|---|
| Pretrained without Policy Finetuning | 75.00 | 87.50 |
| Pretrained with Finetuning | 98.90 | 99.45 |
| RL Baseline | 3.00 | 51.50 |
| No Act Baseline | 0 | 50.00 |
| Random Act Baseline | 0.7 | 50.35 |

Table 9 shows these results. What we see is that indeed, the actions taken by the baselines are not able to verify the hypothesis. The Baseline RL policy only allows the oracle predictor to predict the hypothesis 3% of the time, giving us a upper bound of 51.5% on hypothesis accuracy. Random action is even worse, only leading to the right state sequence 0.7% of the time. No action (the agent that just tries to answer right away) as expected is never able to get the right sequence. For the pre-trainined methods we see that we are able to get to the right states most of the time. The finetuned policy gets the right states almost 100% of the time. With the fixed policy from pretraining, the oracle can answer 75% of the time, meaning that by guessing you could theoretically get to about 88%.

These experiments suggest that as we expected, the hard part of this problem is simultanously learning the policy and prediction is the difficult part. Once you have the best possible hypothesis prediction, RL can quite easily find the correct policy.

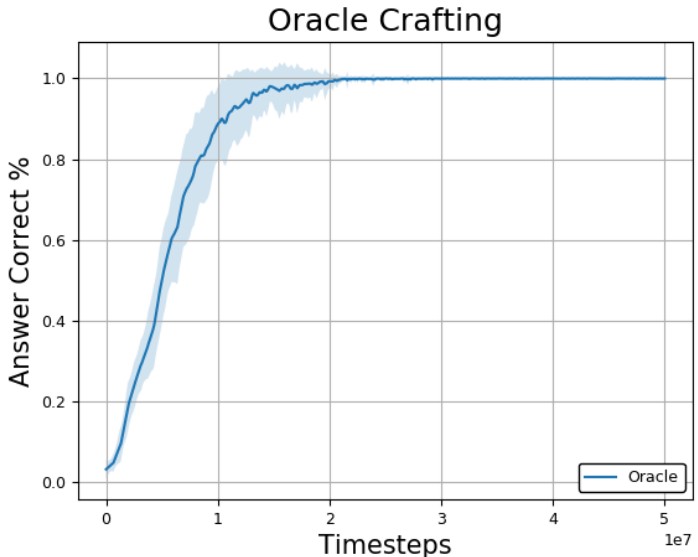

Figure 17: Results on training RL with oracle predictor on crafting environment. Showing mean and variance on 25 random seeds.

# H   LONGER TRAINING BASELINES

Because the pre-trained methods had the benefit of more training frames, we run the baselines for more frames to see whether additional training helps the comparison. We keep all the training parameters the same.

In Figure 18 we show the baseline methods on the original 5 seeds trained for the equivalent 1.5e8 steps. In Figure 19 we show 20 additional seeds trained for longer, although not quite to the 1.5e8 steps.

On the original seeds, training for longer has no effect. However, when we train with many more seeds, we find that for pushblock, we are able to find a random seeds that can get to about 75% accuracy. This is the simplest environment, so it makes sense that this would be the one where the baseline RL might be able to find a policy. The max of this still slightly underperforms the pre-trained policies.

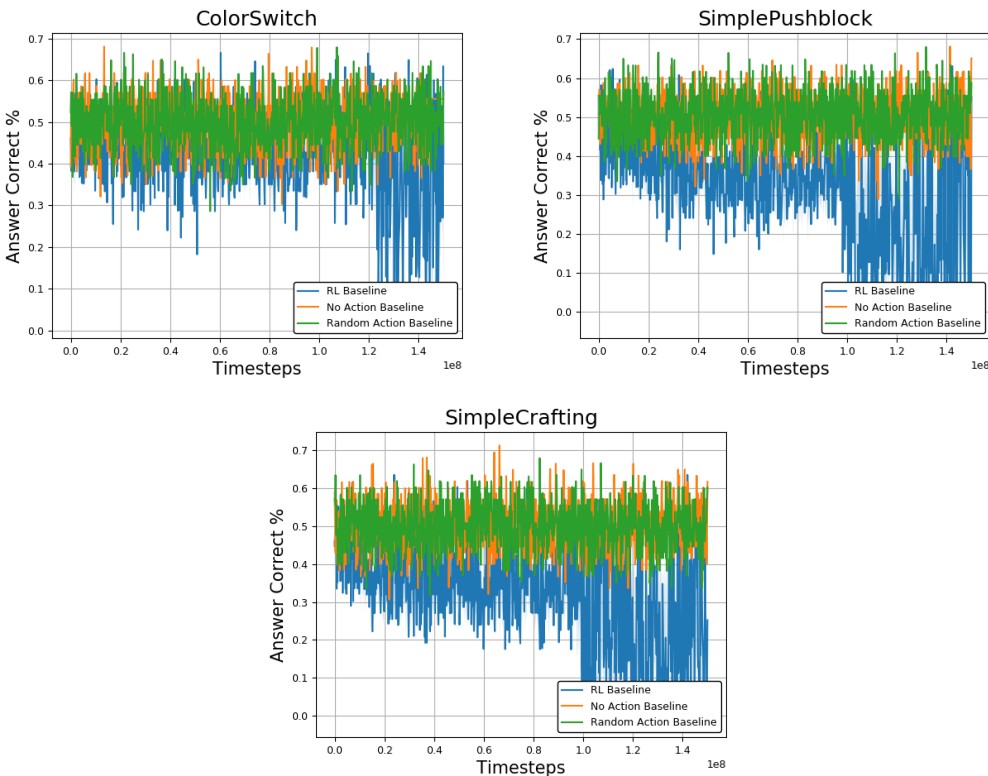

Figure 18: Final hypothesis accuracies of baselines when trained longer. Showing max and variance bands on original 5 seeds.

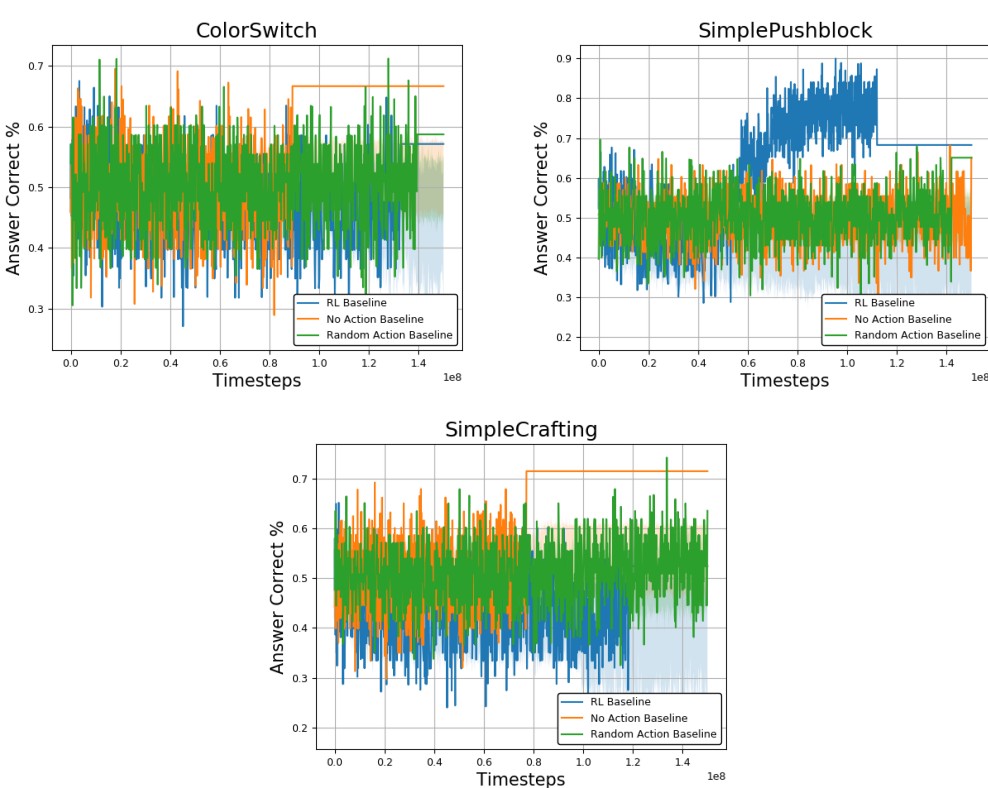

Figure 19: Final hypothesis accuracies of baselines when trained longer. Showing max and variance bands on 25 seeds.

# I MORE STATE MEMORY BASELINES

In this experiment, we see if the RL baseline gets any benefit from increasing $N$, the number of past states it keeps in its observation. We show results for $N = 10, 20, 50, 100$ keeping all other parameters the same.

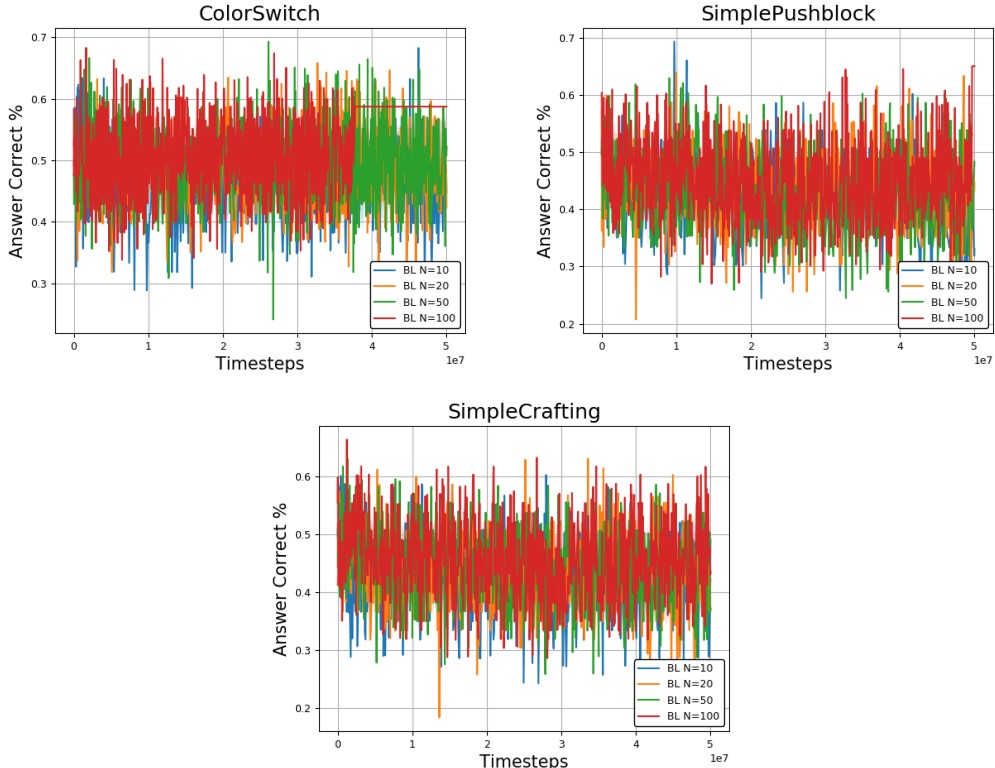

Figure 20: Final hypothesis accuracies of baselines using longer state memory parameter $N$. Showing max and variance. Result on 25 seeds.

We can see that increasing the value of $N$ does not appear to have any effect on the baselines. $N = 5$ is likely sufficient to see the change in the state of the environment and to allow the agent to know to stop and answer the question.

# J    ADDITIONAL RANDOM SEEDS

In this experiment, we show the results from previous experiments, but increase the number of random seeds from the original 5 to 25. When we did this, we also ran 25 random seeds for pretraining, so each results encorporating finetuning came from a different pretraining seed. Results are in Figure 21.

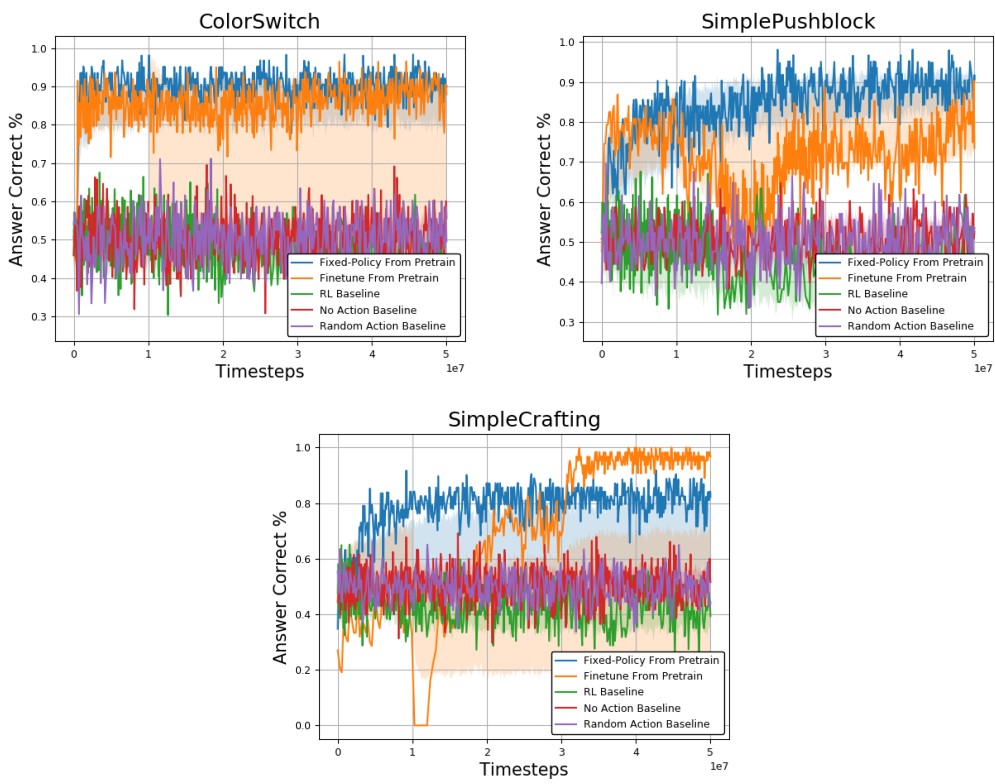

Figure 21: Final hypothesis accuracies of all methods trained with 25 random seeds. Showing max and variance.

Adding more random seeds, we find essentially the same story as with 5 seeds. Finetuned from pretrain is able to get the best single results, but tends to be very high variance. Non-finetuned from pre-train does generally well on everything, except underperformance on crafting (especially on the new templates). And the baselines do still not do well.

One difference worth noting that in Figure 19, we find that training the RL baseline for longer and given more random seeds, it is able to get one good random seed on pushblock. As we noted there, this is the simplest environment, so it makes sense that this would be the one where the baseline RL might be able to find a policy.

For additional clarity, we show these same plots again in Figure 22 with the mean plotted instead of max. This shows the high variance a bit clearer but does not show that we are able to get some good seeds. Appendix E provides a possible solution to this problem by selecting the good random seeds based on a smaller set of hypotheses.

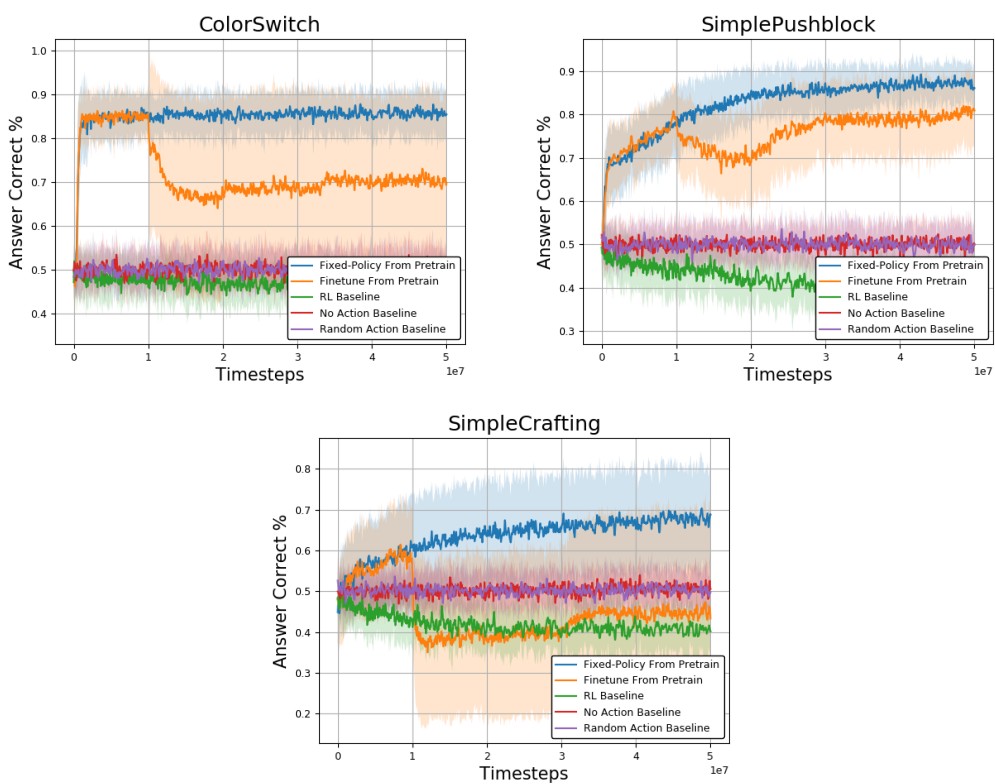

Figure 22: Final hypothesis accuracies of all methods trained with 25 random seeds. Showing mean and variance.

