# OpenReview forum: "Agent as Scientist: Learning to Verify Hypotheses"
_ICLR.cc/2020/Conference — Reject_

### Official Review · AnonReviewer2 · 2019-10-23
**Official Blind Review #2**

**Rating:** 1

**Review:**

The authors present a framework for testing a set of structured hypotheses about environment dynamics by learning an exploratory policy using Reinforcement Learning and a evaluator through supervised learning. They propose a formulation that decomposes environment hypotheses into sets of pre-conditions, required actions, and post-conditions. They then exploit this decomposition to (a) decouple the problem into both RL and supervised learning, and (b) provide localised pre-training to make the problem more tractable.

Overall, I really wanted to like this paper. The problem is interesting, and it certainly provides a great venue for interesting and impactful research in RL, language-conditioned decision making, structured / symbolic learning, and so on. However, I've found it relatively difficult to understand good parts of the methods and part of the experimental section, due to missing or misleading details.

In particular:

1. the justification for splitting the problem in a _exploratory_ / verification policy and a predictor is sound in principle, however it's unclear to me whether the problem is after all that intractable. In the experiment section a "RL Baseline" is mentioned in principle, however (1) it is unclear whether it was pre-trained similarly to the proposed methods, and (2) if the policy has learnt enough about the problems that its poking methodology provides enough signal to the predictor, I would expect the same policy to be able to learn the same function given enough memory and training steps.

2. I'm confused by the way the authors decomposed the action space for the policy and the predictor in section 3.1. Does the policy use ans_T and ans_F at any point during training? Does the actor effectively decide (i.e. by choosing "ans") when to query the prediction network?

3. The way the authors split the templates is confusing to me. Up to section 3.3.1 (and - really - until I read the appendix...), the writing sort of led me to assume that (1) the "(pre-condition, action sequence) -> post-condition" split was a fairly standard manner of compose a hypothesis, and that (2) the templates were mostly symbolic. However after reading the appendix, I found the imposed structure to be fairly arbitrary, and the usage of natural language overkill and not necessarily well justified. Ideally, I would like to see some comparisons between this type of hypothesis and other decompositions used in previous literature, since it seems like the method exploits this particular structure quite heavily and I don't quite understand how it generalises to other tasks.

4. The environments seem to be all fairly similar, both in terms of overall complexity, size, and features. It would have been better to also present problems with fairly different settings (e.g. much different - sparser and/or denser - types of reward function), rather than evaluating multiple times on effectively the same grid-world. I was though encouraged to see that one of the environment seemed to require slightly different setting in the pre-training reward setup, however the authors didn't follow up with some analysis on why there was such a difference.

5. I'm confused by how the pre-training is done. I understand that R_{pre} is used by itself in one environment, but I couldn't figure out whether it's both reward functions at the same time that are used in the rest of them, or just R_{ppost}. Looking at the scale of the (average?) reward, the former seems to be the case, but it would be good to be certain about such things.

6. The final accuracy of all the experiments are shown using the max of top-5, however appendix D shows quite a significant variance for the methods. Thus I'm not sure the analysis and final considerations are reasonable. What happens if the methods are trained on more seeds?

6. [nit] the title is somewhat misleading: in the introduction, a scientist is defined as being both a proposer and a verifier of hypotheses, which is a reasonable, however the authors fundamentally propose to solve only arguably the more straightforward of the two problems. A less _flashy_ title would go a long way towards providing reasonable expectations for the reader.


To improve this paper, I would like to see:

- Better clarity on how the hypothesis setup stands to previous literature.
- The difference in performance on each environment with different pre-training reward function (only one in show in the paper right now)
- At least one more environments with significantly different dynamics, or an explanation of how the existing settings differ in qualitative terms.
- A baseline employing some form of memory (such as heavy usage of frame stacking or recurrency), to attempt at figuring out whether it's really not reasonable to learn the whole problem simply using RL, with ablation of pre-training (which I suspect might make a significant difference).

At this point, I cannot recommend the article for acceptance, but I'd be willing to change my rating if the authors were to address some of the above points.

**Experience Assessment:**

I have published one or two papers in this area.

**Review Assessment: Checking Correctness Of Derivations And Theory:**

I carefully checked the derivations and theory.

**Review Assessment: Checking Correctness Of Experiments:**

I carefully checked the experiments.

**Review Assessment: Thoroughness In Paper Reading:**

I read the paper thoroughly.

---

> ### Author Response · Authors · 2019-11-15
> **Response to Reviewer 2**
>
> Thank you for your helpful comments and suggestions. We have updated the paper and made some general comments above. We will now answer your specific concerns and suggestions.
>
> “The environments seem to be all fairly similar…would have been better to also present problems with fairly different settings (e.g. much different - sparser and/or denser - types of reward function)”:
> We agree and have now included new experiments that explore different forms of pre-training (something that was possible to do in the rebuttal period), which adds to the diversity of experiments presented. More generally, we are working to demonstrate our approach on a broader set of environments, which we hope to include in a future update of the paper.
>
> “a "RL Baseline" is mentioned in principle, however (1) it is unclear whether it was pre-trained similarly to the proposed methods”:
> Pre-training was not used for the RL baseline, but the deep network structure and fine-tuning procedures were the same as our approach.
>
> … “I would expect the same policy to be able to learn the same function given enough memory and training steps.”:
> “A baseline employing some form of memory (such as heavy usage of frame stacking or recurrency), to attempt at figuring out whether it's really not reasonable to learn the whole problem simply using RL”
> Our revised paper includes new experiments that ran the RL baseline for longer (Appendix H), and another experiment that gave the RL baseline more “memory” (Appendix I). These show that the extra training did not help, except for the pushblock task where some limited performance was achieved, although still not as much as our methods. See those appendices for figures and more analysis.
>
> For clarity, all methods already keep a state memory. In all original experiments, the policy and hypothesis predictor get the last N=5 states. In Appendix I we show that increasing N furtehr does not improve performance.
>
> Also, as mentioned to R1, and as we show in Appendix G: when we have an oracle hypothesis verification predictor, the RL baseline with its stacked observations can pretty easily solve the problem. So we don’t believe that the use of memory stacking to sidestep the partial observability is the main difficulty of this problem.
>
> “I'm confused by the way the authors decomposed the action space for the policy and the predictor in section 3.1. Does the policy use ans_T and ans_F at any point during training? Does the actor effectively decide (i.e. by choosing "ans") when to query the prediction network?”
> We apologize for the confusion. The policy does not use ans_T and ans_F, it has a single ans action which then lets the actor query the prediction network. We will make this more clear on revision.
>
> “Better clarity on how the hypothesis setup stands to previous literature”...
> “Ideally, I would like to see some comparisons between this type of hypothesis and other decompositions used in previous literature, since it seems like the method exploits this particular structure quite heavily”:
> As far as we are aware, ours is the first attempt to decompose a hypothesis in this fashion for the purposes of facilitating ML based approach, thus there is no previous literature with which a comparison can be made. However, if R2 knows of any relevant works we would be most interested.
>
> As requested by R3, we have added new experiments to the paper that explores different forms of decomposition, by adjusting the type of intrinsic motivation used. These experiments that the original form of factorization chosen is more effective than other types. See Appendix F for figures and details.
>
> “the usage of natural language overkill and not necessarily well justified”:
> We agree that for the experiments reported, we could have used simpler symbolic representations.  It is not clear that these would actually be more convenient, and they would be less general.  More importantly, we choose templated language because we believe it is scalable in the sense that it allows extension to more complicated and richer hypotheses.  In our view the combinatorial nature of the hypotheses is an important feature of our framing.
>
> “confused by how the pre-training is done”:
> Apologies for the confusion. We will make this clearer in the next revision. When we use both pre and prepost reward functions (which we do on pushblock and crafting) they are added. Your interpretation here was indeed correct.

---

> > ### Author Response · Authors · 2019-11-15
> > **Response to Reviewer 2 (cont)**
> >
> > “accuracy of all the experiments are shown using the max of top-5, however appendix D shows quite a significant variance for the methods”:
> > Yes, this is correct.
> > To address this concern, we performed the experiments described in Appendix E which we hope you will take a look at.  There we show that the selection of the good seeds can be done automatically using our pre-training templates to select good seeds:  that is, the performance on the pre-training templates can be used to reliably pick seeds/models that will result in high performance after fine-tuning. Thus even though the variance is high, bad runs can be weeded out in the first phase with the pre-training templates.
> >
> > “what happens if the methods are trained on more seeds?”:
> > In Appendix J we rerun the final accuracy experiments using more random seeds. We also are sure in the other experiments (including our method of random seed selection in Appendix E) to run with more random seeds.
> >
> > This does not appear to substantially change the results, at least for this experiment. However, in Appendix H, when we run more random seeds AND train the baselines for longer, the pushblock RL baseline gets one good random seed, although it still performs worse than our methods (~75% versus 85%). No amount of additional training or random seeds gives good results for baselines on the other environments.
> >
> > [Would like an experiment realted to] the difference in performance on each environment with different pre-training reward function (only one in show in the paper right now):
> > As we said above, as requested by R3, we have added new experiments to the paper that explores different ways of doing pre-training. See Appendix F for figures and details.
> >
> > “the title is somewhat misleading…”:
> > We would like to retain “Scientist” in the title, since it conveys the ultimate goal of this line of work. That said, we understand your concern. We therefore intend to change the title to: “Toward a Scientist Agent: Learning to verify hypotheses”

---

### Official Review · AnonReviewer3 · 2019-10-23
**Official Blind Review #3**

**Rating:** 3

**Review:**

This paper trains agents which are able to verify hypotheses, such as “the blue switch causes the door to open”. It does this by first pretraining the agent to perform interventions in the environment which change the states of the objects of interest, and then finetuning the agent to actually make a decision about whether the given hypothesis is correct. The paper shows that agents trained using this procedure are able to not only verify the types of hypotheses seen during pre-training, but also learn to verify more complex hypotheses. In contrast, an agent which is trained directly on the hypothesis verification task is unable to learn to do it.

Overall, I enjoyed reading this paper and thought that it provided an interesting take on the question of how to train agents that can appropriately gather information about their environments. However, (1) the paper lacks any discussion of related work in terms of causal reasoning and partial observability, and (2) the experiments and analysis seem weak. I thus am giving a score of “weak reject”, though it is possible I could increase my score is some of my concerns can be addressed.

First, I was very surprised to see that the paper included no discussion at all about either causal reasoning or partial observability. The whole notion of verifying hypotheses—particularly those in the triplet form as presented in the paper—is equivalent to the idea of performing inference about the structure of a causal graph with three variables. The choice of which interventions to perform in order to make these inferences is a well-studied problem [1] and has been recently explored in the context of RL as well [2]. The novelty here seems to be in embedding the problem of causal reasoning in harder credit assignment problems (i.e. longer time horizon), though see [3]. Similarly, the setup of the MDP in the paper is actually a POMDP, where the state includes the truth value of the hypothesis but where observations do not include this information. Yet, there is no mention of POMDPs or discussion of the literature on partial observability in the paper.

Second, I felt that the setup was overly complex in places making it difficult to draw conclusions, that there were a lack of comparisons, and that the analysis was not as in depth as it could have been. For example, why is it necessary to represent the hypothesis with natural language? Why not use a symbolic representation? It seems like including the pseudo-natural language adds unnecessary complexity and makes it difficult to distentangle what about the problem is hard (Understanding the hypothesis? Choosing the right interventions? Parsing the observations correctly?). The utility of having it be closer to language is that you might see generalization between related hypotheses, but this isn’t really something that is actually tested for since all hypotheses are trained on either during pretraining or finetuning.

I also feel like the choice of pretraining reward feels somewhat arbitrary, and it would have been nice to see comparisons to other alternatives (and even better, to other forms of intrinsic motivation). For example, here are a few alternate ways of rewarding the agent that seem intuitively like they could also work:
Reward the agent for changing the state of any of the objects in the environment
Reward the agent for changing the state of any object referenced in the hypothesis
Reward the agent for observing a state of the world it has not seen before (i.e. count-based exploration)
In other words, how important is the fact that the reward is given based on the pre and postconditions?

I thought the paper would benefit from more detailed analyses to tease apart the behavior of the agent. For example, I am curious how many errors are a result of errors in the predictor versus poor exploration behavior by the policy. Could you report (1) how frequently the policy’s behavior results in the right observations necessary to make a decision, and (2) results with a policy which uses an oracle predictor (i.e. which will always report the correct answer, if there was enough data in the last N frames to detect that answer)?

On the more practical side, I also thought the quality of the evaluations was not very thorough. For instance, it looks like the pretraining proceeds for 1e8 steps and finetuning for 5e7 steps, based on the plots (these values should be stated more explicitly in the paper). However, this is a bit of an unfair comparison for the “RL Baseline”, as it only is trained for 5e7 steps while the other agents are trained for 1.5e8 steps. I would like to see a comparison where the RL Baseline agent is trained for 1.5e8 steps as well. Similarly, on the bottom of page 6 the paper says “we show the max out of five for each of the methods shown”. However, only reporting the max value is considered bad practice and can result in misleading comparisons (see Joelle Pineau’s talk on “Reproducible, Reusable, and Robust Reinforcement Learning” at NeurIPS 2018). I’d like to see the data in all figures and tables reported with means or medians across seeds, rather than best seeds.

A few minor comments:

- Please state in the main text which RL algorithm you use.
- Can you clarify whether Figure 2 show the proxy rewards or the true rewards?
- For R_pre and R_ppost, what values do you use for C and N?

[1] Pearl, J. (2000). Causality: models, reasoning and inference (Vol. 29). Cambridge: MIT press.
[2] Dasgupta, I., Wang, J., Chiappa, S., Mitrovic, J., Ortega, P., Raposo, D., ... & Kurth-Nelson, Z. (2019). Causal reasoning from meta-reinforcement learning. arXiv preprint arXiv:1901.08162.
[3] Denil, M., Agrawal, P., Kulkarni, T. D., Erez, T., Battaglia, P., & de Freitas, N. (2016). Learning to perform physics experiments via deep reinforcement learning. arXiv preprint arXiv:1611.01843.

--
Update after rebuttal:

Thank you very much for your response. However, I do not feel that all of my concerns have been addressed and thus will keep my score as it is. In particular, I still feel the paper lacks sufficient discussion of the literature on causal reasoning. I also do not think it is sufficient to add an appendix with the results across multiple seeds: these results should be in the main paper. I'm not sure I follow the justification that max seeds make sense because "the reward distribution is quite binary in nature"---the plots shown in Figure 3 and 4, for example, span a range of values from 0 to 1. I find the plots that have both variance and max seed very hard to interpret---in some cases the mean is so much lower than the max seed that the variance region doesn't overlap at all. More broadly, it might be easier to compare using bar plots showing final performance, rather than training curves

I appreciate the additional results, especially with different pretraining schemes---thanks for adding these! I have a bit of hard time interpreting the results though since there are no direct comparisons with the triplet pretraining scheme; it would be helpful if these results could be included in these figures too.

**Experience Assessment:**

I have read many papers in this area.

**Review Assessment: Checking Correctness Of Derivations And Theory:**

N/A

**Review Assessment: Checking Correctness Of Experiments:**

I assessed the sensibility of the experiments.

**Review Assessment: Thoroughness In Paper Reading:**

I read the paper at least twice and used my best judgement in assessing the paper.

---

> ### Author Response · Authors · 2019-11-15
> **Response to Reviewer 3**
>
> Thank you for your helpful comments and suggestions. We have updated the paper and made some general comments above. We will now answer your specific concerns and suggestions.
>
> “the paper included no discussion at all about either causal reasoning or partial observability”:
> Yes, you are correct; the setting is POMDP, not an MDP. We have fixed this in the text. All models (including all baselines) were provided with a history of observations. Also, not contextualizing our work with the papers you suggest was an omission on our part that has been remedied in the updated version. Note that https://arxiv.org/pdf/1611.01843.pdf also takes the strategy of using standard RL with models that use a history of observations (e.g. RNN).
>
> “why is it necessary to represent the hypothesis with natural language? Why not use a symbolic representation?”
> As remarked to R2: we agree that for the experiments reported, we could have used simpler symbolic representations. It is not clear that these would actually be more convenient, and they would be less general. More importantly, we choose templated language because we believe it is scalable in the sense that it allows extension to more complicated and richer hypotheses.  In our view the combinatorial nature of the hypotheses is an important feature of our framing.
>
> “only reporting the max value is considered bad practice ...I’d like to see the data in all figures and tables reported with means or medians across seeds, rather than best seeds”:
> See Appendix J for the main result (with more random seeds) with the mean and variance.
>
> However, the reward distribution is quite binary in nature (either it works, or fails completely) thus neither mean, median or max measures provide an adequate characterization. However, we are able to side-step this issue by using our pre-training mechanism to select good seeds: in a new experiment.
>
> To address this concern, we performed the methodology in Appendix E which we hope you will take a look at. Here we show that the selection of the good seeds can be done automatically using our pre-training templates to select good seeds: in a new experiment we show that the performance on the pre-training templates can be used to reliably pick seeds/models that will result in high performance after fine-tuning.
>
> “Could you report (1) how frequently the policy’s behavior results in the right observations necessary to make a decision, and (2) results with a policy which uses an oracle predictor”:
> This was a very good suggestion. We did this analysis and experiments in Appendix G for the crafting environment (in the interest of time, we did not have time to do more). We will provide this for all environments on revision.
>
> “I would like to see a comparison where the RL Baseline agent is trained for 1.5e8 steps as well.”
> Another good point and suggestion. We provide this in appendix H.
>
> “alternate ways of rewarding the agent”
> This was another good suggestion, and we really appreciate this suggestion.
> We followed two of these suggestions:
> 1. Reward the agent for changing the state of any of the objects in the environment
> 2. Reward the agent for changing the state of any object referenced in the hypothesis
>
> We didn’t do the count-based exploration for a couple reasons: (time and because even in 5x5 gridworlds, there are an exponential number of states and we don’t believe that this would have done very well).
>
> This experiment is in Appendix F. We provide details and analysis there.
>
> To answer your question though: “how important is the fact that the reward is given based on the pre and postconditions?”
> It’s pretty important.
> Some versions of intrinsic motivation you suggest do pretty well, especially the crafting one because it turns out to be a pretty good proxy for how to verify the hypothesis, but much less well on color switch and pushblock. See the paper for more analysis here.
>
> Minor points:
> “which RL algorithm you use”:
> PPO
> “whether Figure 2 show the proxy rewards or the true rewards?”
> Proxy rewards.
> “For R_pre and R_ppost, what values do you use for C and N?”
> C=10, N=5

---

### Official Review · AnonReviewer1 · 2019-10-25
**Official Blind Review #1**

**Rating:** 3

**Review:**

The paper looks into the problem of training agents that can interact with their environments to verify hypotheses about it. It first formulates the problem as a MDP, where the agent takes actions to explore the environment and has two special actions (Answer_True, and Answer_False) to indicate that the agent has made a prediction about the validity of the hypothesis. The reward depends on how correct the agent's prediction is. A second formulation uses MDP to explore the states and has a special action (Answer), which predicts the validity of the hypothesis based on the last sequence of N states visited. This is one side of the problem. The authors carry out such experiments and conclude that this doesn't work.

Then, the authors exploit the structure of some hypotheses (such as triplet hypotheses of the form pre_condition, action_sequence, post_condition), which are easier to test. They conclude that taking this structure into account helps.

Overall, the paper is well-written and the literature review section is quite excellent. However, I have reservations against the formulations that the authors used. I would appreciate it if the authors present their argument in the rebuttal.

First, in the plain formulation of MDP, a policy produces an action according to the current state only. The authors add (Answer_True, and Answer_False) to the list of actions in MDP. So, if the agent is trained on some hypotheses, the agent will essentially learn to identify for each h which state s that can be used to to verify h (either prove or disprove it). To me, this is essentially memorization, and the agent cannot learn to predict the validity of new hypotheses. So, it seems that formulating the problem using MDP is not reasonable to begin with.

Second, when the agent exploits the structure of the hypotheses, the problem becomes nearly trivial. It would have been interesting if, somehow, the agent learned the strategy of trying to alter the preconditions or postconditions on its own, but this is not the case in the paper. The formulation essentially tells the agent that it should alter the preconditions and postconditions so that we have enough information about the validity of h that can be fed into a prediction network. I think that the fact this works is not that interesting.

Some minor comments:
- I suggest that all acronyms be defined in the paper before they are used.
- In the reward functions, why did the authors use C instead of just using 1.
- In Page 4, "The agent is is spawned" has a typo.
- In Page 5, "so we can in principal only" has a typo.
- In Page 7, "as it paves the towards" has a typo.

==================
#Post Rebuttal Remark

I have gone through the authors' response and I thank them for it, particularly for making some of the suggested enhancements. However, my score remains unchanged.


**Experience Assessment:**

I do not know much about this area.

**Review Assessment: Checking Correctness Of Derivations And Theory:**

N/A

**Review Assessment: Checking Correctness Of Experiments:**

I assessed the sensibility of the experiments.

**Review Assessment: Thoroughness In Paper Reading:**

I read the paper at least twice and used my best judgement in assessing the paper.

---

> ### Author Response · Authors · 2019-11-15
> **Response to Reviewer 1**
>
> Thank you for your helpful comments and suggestions. We have updated the paper and made some general comments above. We will now answer your specific concerns and suggestions.
>
> “when the agent exploits the structure of the hypotheses, the problem becomes nearly trivial...”:
> We agree that the structure imposed during pre-training makes the problem far simpler than the original one (but not trivial -- our RL approach still only succeeds intermittently). But rather than a weakness, it should be viewed as a strength of the way we have chosen to frame the problem. As noted, the original problem of hypothesis verification is just too hard by itself (RL baseline makes little progress at all). Our idea of decomposing the hypothesis into (pre,post,reward) is an important contribution since it makes the problem amenable to ML approaches. Importantly, once pre-trained using this assistance, the model can be fine-tuned on the original task, i.e. without the need to use the imposed structure. Hence the use of our imposed structure as a pre-training step makes a hitherto inaccessible problem tractable.
>
> R3 requested an exploration of other forms of structure, imposed via different intrinsic forms of motivation to the one in the paper. We have followed these suggestions, revising the paper to include these new experiments (see Appendix F). These show that alternate structures (which provide weaker assistance), are in general not sufficient to make the problem tractable, lending support to our original formulation.
>
> “essentially memorization...cannot learn to predict the validity of new hypotheses”:
> Our approach is clearly not memorizing: (i) the environment is randomized (locations of agents, objects, hypotheses and ground truth) each episode giving rise to an exponential number of states and (ii) in Table 1, we demonstrate the learning is able to generalize to new templates.
>
> “First, in the plain formulation of MDP, a policy produces an action according to the current state only.” …. “at formulating the problem using MDP is not reasonable to begin with”:
>
> We are sorry for the confusion- we incorrectly wrote that the problem is an MDP, we should have written POMDP; see also our response to reviewer 3.
>
> There are quite a few works which do frame stacking, or use a recurrent network structure as a way of using RL on problems which may require more than the current observation frame. Frame stacking is very commonly used in Atari for instance (Mnih, Volodymyr, et al. "Playing Atari with Deep Reinforcement Learning." arXiv preprint arXiv:1312.5602 (2013).) Lots of works, such as the previously cited (Das et al., 2018) keep this history implicitly using RNNs.
>
> As we show in Appendix G, when we have an oracle hypothesis verification predictor, the RL baseline with its stacked observations can pretty easily solve the problem. So we don’t believe that the use of memory stacking to sidestep the partial observability is the main difficulty of this problem.
>
> “Typos, undefined acronyms”
> Thank you. Those typos have been fixed. We will be sure to read carefully before the next revision to find more typos and to define all our acronyms.

---

### Author Response · Authors · 2019-11-15
**Response to all reviewers**

We would like to express our sincere thanks to reviewers, whose feedback is among the most constructive, useful and detailed that we’ve ever received at ICLR and similar venues. Your collective input has caused us to make many changes to the paper, in terms of writing as well as new experiments that result in what we believe to be greatly strengthened work.   We are pleased to report that we have done *ALL* the experiments that you asked for (except running on completely new environment which would require substantial more time and experiments). These have all been added to the paper as appendices, but can be moved to the main body of the paper later (subject to space constraints).

The experiments include:
A different way of pruning and analyzing the variance of our methods by using the triplet hypotheses as a sort of validation set to choose random seeds to address concerns about variance and displaying max results. (Appendix E).
Experiments using different pre-training reward functions suggested by one of the reviewers. (Appendix F)
An experiment training an RL agent using an oracle hypothesis predictor, and an evaluation of our methods using this oracle to see where our methods and baselines do well or badly. (Appendix G)
Experiments where we train our baseline methods for longer to account for the baselines not receiving the pre-training time (Appendix H).
An experiment seeing whether giving the baseline a longer state memory improves performance (Appendix I)
Rerunning our results on more random seeds to see if there is a substantial change in the results (Appendix J)

The goal of this work is to train agents that can learn to verify different kinds of hypotheses. In our view, the key contribution of this work is showing that the structure of the hypothesis verification problem gives an opening for solving it, and that the problem is difficult without using the particular structure. The reviewers’ comments and the results of their suggested experiments strengthen this conclusion.

Given the detailed nature of the comments, we will respond separately to each reviewer separately on their specific concerns. We encourage all reviewers to look at the new experiments.

---

### Decision · Program_Chairs · 2019-12-19

**Decision:**

Reject

**Comment:**

The authors propose an agent that can act in an RL environment to verify hypotheses about it, using hypotheses formulated as triplets of pre-condition, action sequence, and post-condition variables. Training then proceeds in multiple stages, including a pretraining phase using a reward function that encourages the agent to learn the hypothesis triplets.

Strengths: Reviewers generally agreed it’s an important problem and interesting approach

Weaknesses: There were some points of convergence among reviewer comments: lack of connection to existing literature (ie to causal reasoning and POMDPs), and concerns about the robustness of the results (which were only reporting the max seeds).  Two reviewers also found the use of natural language to unnecessarily complicate their setup. Overall, clarity seemed to be an issue. Other comments concerned lack of comparisons, analyses, and suggestions for alternate methods of rewarding the agent (to improve understandability).

The authors deserve credit for their responsiveness to reviewer comments and for the considerable amount of additional work done in the rebuttal period. However, these efforts ultimately didn’t satisfy the reviewers enough to change their scores. Although I find that the additional experiments and revisions have significantly strengthened the paper, I don't believe it's currently ready for publication at ICLR. I urge the authors to focus on clearly presenting and integrating these new results in a future submission, which I look forward to.